# Estimating The Sea Ice Floe Size Distribution Using Satellite Altimetry: Theory, Climatology, and Model Comparison

Christopher Horvat[a], Lettie A. Roach[b,c], Rachel Tilling[d,e], Cecilia M. Bitz[b], Baylor Fox-Kemper[a], Colin Guider[f], Kaitlin Hill[g], Andy Ridout[h], and Andrew Shepherd[i]

[a]Institute at Brown for Environment and Society, Brown University, Providence, RI, USA
[b]Department of Atmospheric Sciences, University of Washington, Seattle, WA, USA
[c]National Institute of Water and Atmospheric Research, Wellington, NZ
[d]Cryospheric Sciences Laboratory, NASA Goddard Space Flight Center, Greenbelt, MD, USA
[e]Earth System Science Interdisciplinary Center, University of Maryland, College Park, MD, USA
[f]Department of Mathematics, University of North Carolina, Chapel Hill, NC, USA
[g]School of Mathematics, University of Minnesota, Minneapolis, MN, USA
[h]Centre for Polar Observation and Modelling, University College London, London, UK
[i]Centre for Polar Observation and Modelling, University of Leeds, Leeds, UK

**Correspondence:** Christopher Horvat (horvat@brown.edu)

**Abstract.** In sea-ice-covered areas, the sea ice floe size distribution (FSD) plays an important role in many processes affecting the coupled sea-ice-ocean-atmosphere system. Observations of the FSD are sparse — traditionally taken via a pain-staking analysis of ice surface photography — and the seasonal and inter-annual evolution of floe size regionally and globally is largely unknown. Frequently, measured FSDs are assessed using a single number, the scaling exponent of the closest power law fit to the observed floe size data, although in the absence of adequate datasets there have been limited tests of this "power-law hypothesis". Here we derive and explain a mathematical technique for deriving statistics of the sea ice FSD from polar-orbiting altimeters, satellites with sub-daily return times to polar regions with high along-track resolutions. Applied to the CryoSat-2 radar altimetric record, covering the period from 2010-2018, and incorporating 11 million individual floe samples, we produce the first pan-Arctic climatology and seasonal cycle of sea ice floe size statistics. We then perform the first pan-Arctic test of the power law hypothesis, finding limited support in the range of floe sizes typically analyzed in photographic observational studies. We compare the seasonal variability in observed floe size to fully coupled climate model simulations including a prognostic floe size and thickness distribution and coupled wave model, finding good agreement in regions where modeled ocean surface waves cause sea ice fracture.

## 1   Introduction

Earth's polar oceans are covered with sea ice: a thin, heterogeneous interface that plays an important role in the coupling between ocean and atmosphere. Sea ice is a collection of many individual pieces, called floes, which may be characterized in terms of a horizontal length scale, their "size". On the large scales relevant to global climate modeling, the statistical variability of floe size is described using the floe size distribution (FSD,  Rothrock and Thorndike, 1984).

The FSD is an important property of the sea ice cover that influences the multiscale temporal and geographic variability of sea ice, akin to the grain size in sedimentology or particle size distribution in atmospheric chemistry. The scale of individual floes plays a role in many sea-ice-related processes: sea ice melt rate (Steele, 1992; Horvat et al., 2016; Horvat and Tziperman, 2018), the evolution of the oceanic mixed layer (Manucharyan and Thompson, 2017), atmospheric boundary layer exchange (Birnbaum and Lüpkes, 2002; Lüpkes and Birnbaum, 2005; Tsamados et al., 2014), the sea ice response to applied stress (Feltham, 2008; Wilchinsky and Feltham, 2011), and the propagation of waves into the ice (Squire et al., 1995; Squire, 2007; Smith and Thomson, 2016). The importance of the sea ice FSD has led to the development of diagnostic FSD models of varying complexity (Williams et al., 2013; Zhang et al., 2015; Bateson et al., 2019), and a prognostic floe size and thickness distribution (FSTD) scheme (Horvat and Tziperman, 2015; Roach et al., 2018a).

Despite the potential relevance of sea ice floe size to polar climate evolution, there remain no climate-scale assessments of average floe size or the FSD. The observational record of floe statistics derives from visual imagery localized in space and time (i.e., Rothrock and Thorndike, 1984; Toyota et al., 2006; Steer et al., 2008; Toyota et al., 2011) or from repeat measurements in the same region over multiple months (Hwang et al., 2017; Stern et al., 2018a), which may subsequently used to compile a seasonal cycle of the FSD (Perovich and Jones, 2014; Stern et al., 2018a). FSD measurements are obtained by identifying individual floes within a 2-dimensional image of the sea-ice surface. Because floe sizes span several orders of magnitude, accurate representations of the FSD — even in relatively small geographical domains and in perfect lighting and surface conditions — require high resolution and high observational coverage. Nearly all measurements of the FSD have been made in accordance with a "power law" scaling hypothesis commonly used to describe multiscale systems (Mandelbrot and Wheeler, 1983), in which the resulting FSD is fit to a straight line in logarithmic coordinates, whose slope, $\alpha$, is reported as an intrinsic property of the floe mosaic. There is large uncertainty in these scaling coefficients, the range they apply over, and their applicability and origin (Herman, 2011; Horvat and Tziperman, 2017; Herman et al., 2018; Stern et al., 2018b). Improvements in the quality and quantity of available FSD data are needed before arriving at consensus derived FSD statistics to guide and assess model performance.

Here we outline a method that exploits satellite radar altimetry to construct the FSD and its moments across polar regions with sub-kilometer spatial resolution, sub-daily temporal resolution, and spanning multiple orders of magnitude in size. Altimeters, like the ones carried on the Envisat, ICESat, CryoSat-2, and ICESat-2 satellites, make repeated, frequent passes over polar oceans, and substantial efforts have been made to process the satellite returns to discriminate between open water, floes, and leads. The altimetric returns have found many uses, including reconstructing the sea ice thickness field (Laxon et al., 2013; Tilling et al., 2016, 2018b) and ocean surface circulation under sea ice (Peacock and Laxon, 2004; Armitage et al., 2018). Fields inferred from altimetry have led to advances in understanding polar systems: from forecast and climate prediction (Day et al., 2014) to model validation (Schröder et al., 2018; Allard et al., 2018) to climate change studies (Laxon et al., 2003; Kwok, 2018), and have been evaluated and validated using field campaign data (Skourup et al., 2017; Sandberg Sorensen et al., 2018; Tilling et al., 2018b).

One-dimensional measurements of sea ice properties, like along-track altimetric measurements of ice open water, have long been sought to describe the two-dimensional ice surface. Rothrock and Thorndike (1984) originally described a method for

reconstructing the sea ice floe size distribution in a region using straight-line measurements over the geometry of floes. Lindsay and Rothrock (1995) later compiled the statistics of lead and ice spacings in two-dimensional imagery. Other work has taken place to derive and understanding the width distribution of individual leads in visual imagery and altimetry (Wadhams et al., 1988; Key and Peckham, 1991; Key, 1993; Wernecke and Kaleschke, 2015), which can be used to estimating heat fluxes and turbulent transfer between the ocean and atmosphere. To date, however, these studies have not been designed to facilitate a comparison with model data, nor have altimetric studies been used to compile floe size statistics. These objectives are the focus of this work.

We outline the mathematical theory that allows for comparing altimetric datasets and the FSD in Sec. 2. In Sec. 3 we apply this method to a new dataset of segmented CryoSat-2 sea ice type data from 2010-2018. Using this data we produce the first climatological maps of mean sea ice floe size and fragmentation for the Arctic Ocean. We then test the power law hypothesis, finding limited support for power-law scaling across most of the dataset in Sec. 4. One of the key aims of the paper is to develop floe size distribution measurements that are useful for model validation and calibration. In Sec. 5, we show a proof-of-concept, demonstrating how altimetric data can be used to constrain and evaluate new models of the FSD, comparing the CryoSat-2 FSD data to a climate model simulation with a prognostic FSTD model. We conclude in Sec. 6.

## 2 Floe Chords and the Floe Size Distribution

For an individual pass over sea ice by a polar-orbiting satellite altimeter, return waveforms along the satellite orbit track are assigned a surface type depending on the waveform shape and coincident sea ice concentration (Tilling et al., 2018b). A "floe chord" of length $D$ is a continuous series of points identified as sea ice, covering a geographic distance $D$ (Tilling et al., 2018a, 2019). Define a floes size, $r$, as its "effective radius" — the square root of the floe's area divided by $\pi$ (Rothrock and Thorndike, 1984; Horvat and Tziperman, 2015)) We use radius instead of diameter, as appears in some other observational studies, for comparison with model output in Sec. 5. Because the satellite path is at an unknown angle with respect to the (also unknown) floe geometry, any individual floe chord measurement is not a floe size measurement. Converting between suitably processed altimetric floe chord measurements and floe size statistics is therefore the subject of this section. Details on the processing of the CryoSat-2 waveform, used to produce a dataset of floe chords spanning the period 2010-2018, is outlined in Sec. 3 and Tilling et al. (2019).

For a domain of horizontal area $A$, and over a period of time $\Delta T$ that corresponds to several repeat satellite passes, we bin the set of recorded floe chords to form a probability distribution $S(D)$, which we term the "floe chord distribution" (FCD), where $S(D)dD$ is equal to the number fraction of floe chords in $A$ over $\Delta T$ with length between $D$ and $D + dD$, and is normalized to one. To collapse all measured chords onto a single independent scalar coordinate ($D$), we follow the example of turbulence statistics (Batchelor, 1953) and assume that the floe chord distribution data is homogeneous, isotropic, and stationary within the region and time data is collected. In the same region, we define the (non-cumulative) number FSD $P(r)$, where $P(r)dr$ is the fractional number of floes with a size between $r$ and $r + dr$ in $A$, and is also normalized to one. The FSD inherits the

assumptions of homogeneity, isotropy, and stationarity from the FCD. Our objective is to relate the FCD, $S(D)$, or quantities derived from the FCD, to the statistics of the FSD, $P(r)$.

Bayes' theorem relates $S(D)$ and $P(r)$ through conditional probabilities,

$$F(r;D)S(D) = \tilde{F}(D;r)P(r). \tag{1}$$

The conditional probability $F(r;D)$ relates given chord lengths to the floe size distribution that could generate them: $F(r;D)dr$ is the probability that floes with size in the range from $r$ to $r+dR$ were sampled given a chord of length $D$. The conditional probability $\tilde{F}(D;r)$ relates given floe sizes to the chord length distribution they generate: $\tilde{F}(D;r)dD$ is the probability of measuring a floe chord of length from $D$ to $D+dD$ given that a floe of size $r$ was measured.

This second probability distribution $\tilde{F}(D;r)$ can be derived from first principles under a single assumption: that the chord
length distribution that would be sampled from a set of floes of size $r$ is independent of $r$ (equivalently, the floe shape distribution is scale-invariant). Formally, this requirement is,

$$\tilde{F}(D;r)dD = G(\xi)d\xi, \tag{2}$$

where $G(\xi) = G(\frac{D}{2r})$ is an unknown function that integrates to 1 over the interval from $\xi = 0$ to 1. Under this assumption, the distribution of possible chord lengths measured from floes of size $r$ has the same functional form independent of $r$. The proba-
bility distribution $F(D;r)$ may be derived by considering the geometric relationship between straight-line satellite passes and the geometry of the floes they pass over. Individual floe shapes are highly variable: making an assumption about the distribution of floe shapes may introduce biases in the statistics derived from the FCD. Yet as we prove in Appendix A, the ability to derive FSD statistics from the FCD does not depend on the precise form of $\tilde{F}(D;r)$ so long as the homogeneous, isotropic, stationary and scale-invariance assumptions are retained, and the evaluation of power-law scaling is in fact independent of $\tilde{F}(D;r)$.

To proceed and arrive at a concrete (although not general) realization of these functions, we will assume all floes are perfect circles. In assessments of the relationship between major and minor axes of individual floes, the "roundness" parameter for a floe is typically within 15% of one (Rothrock and Thorndike, 1984; Toyota et al., 2011; Perovich and Jones, 2014; Gherardi and Lagomarsino, 2015; Alberello et al., 2019), suggesting that this circular assumption, while simplistic, is broadly appropriate. Nevertheless, it will likely be necessary to amend the analysis below in the future to account for more realistic shape
distributions and geometries (e.g., diamonds (Wilchinsky and Feltham, 2006)), regional differences in floe shape properties (such as in regions where shear stress determines fracture patterns and floe shapes (Schulson and Hibler, 1991)), or to evaluate the sensitivity of the results that follow to the assumed shape distribution. Solving for $\tilde{F}(D;r)$ is a geometric problem that relates the possible measured chord lengths to the underlying floe size, and we solve this explicitly for circular floes here. Similar geometric problems have been identified and solved in other fields (e.g., Pons et al., 2006; Nere et al., 2007), and we
therefore leave refinement of $\tilde{F}(D;r)$ to future work.

Consider the special case that all floes are perfect circles, illustrated in Fig. 1. Because there is no correlation between the statistics of local sea ice deformation and pre-determined satellite tracks, an individual recorded floe chord, $D$, originating from a floe of radius $r$, was obtained from a satellite trajectory that crosses the floe at a random interior angle $\theta$, thus the distribution of

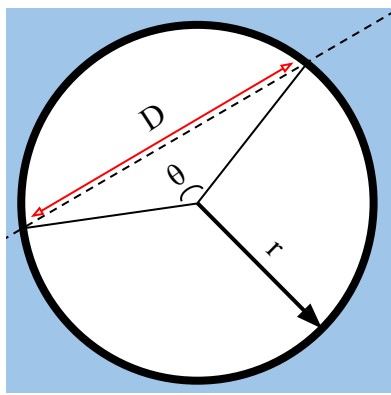

**Figure 1.** Relating a floe chord to floe size for a circular floe. A satellite track (dashed black line) passes over a floe of radius $r$ (solid black line). The track records a series of echoes of length $D$, which is the length of a chord (red line) identified by its interior angle, $\theta$.

$\theta$ is uniform. Because of rotational symmetry, we need only consider $\theta \in [0, \pi)$, sampled according to a probability distribution $T(\theta; r) = \pi^{-1}$. The length $D$ is thus a chord of this circular floe, with $D = 2r \sin(\theta/2)$. Accordingly,

$$\tilde{F}(D; r) = T(\theta; r) \frac{\partial \theta}{\partial D} = \begin{cases} \frac{2}{\pi} \frac{1}{\sqrt{(2r)^2 - D^2}} & r > D/2, \\ 0 & \text{otherwise}, \end{cases} \tag{3}$$

which is a probability function that meets the above criterion (2).

5     The n$^{\text{th}}$ moment of the floe chord distribution $S(D)$, is defined,

$$\langle D^n \rangle \equiv \int_0^\infty D^n S(D) dD = \int_0^\infty dr\, P(r) \int_0^\infty D^n \tilde{F}(D; r) dD. \tag{4}$$

For any function $\tilde{F}(D; r)$ satisfying the scale-invariance above, the right-hand-side may be expressed in terms of moments of $P(r)$ (see Appendix A). For circular floes, using Eq. 3,

$$\langle D^n \rangle = \int_0^\infty dr\, P(r) \int_0^{2r} \frac{2}{\pi} \frac{D^n}{\sqrt{(2r)^2 - D^2}} dD = \int_0^\infty dr\, P(r) \frac{2^{n+1}}{\pi} r^n \int_0^{\frac{\pi}{2}} \sin(x)^n dx = A_n \langle r^n \rangle, \tag{5}$$

10    where $\frac{D}{2r} \equiv \xi = \sin(x)$, $\langle r^n \rangle$ is the n$^{\text{th}}$ moment of $P(r)$ and the coefficient $A_n$ is,

$$A_n \equiv \int_0^1 \xi^n G(\xi) d\xi = \frac{2^{n+1}}{\pi} \int_0^{\frac{\pi}{2}} \sin(x)^n dx = \frac{2^n}{\pi} B\left(\frac{n+1}{2}, \frac{1}{2}\right),$$

where $B$ is the beta function. For $n = 0$, 1, 2, or 3, then $A_n$ is 1, $\frac{4}{\pi}$, 2, or $\frac{32}{3\pi}$. Two important FSD-derived quantities are derived from ratios of FSD moments, and therefore can be obtained from the FCD directly: the "representative radius" (Horvat and

Tziperman, 2017; Roach et al., 2018a),

$$\bar{r} \equiv \frac{\int\limits_0^\infty r^3 P(r)dr}{\int\limits_0^\infty r^2 P(r)dr} = \frac{\langle r^3 \rangle}{\langle r^2 \rangle} = \frac{3\pi}{16}\frac{\langle D^3 \rangle}{\langle D^2 \rangle}. \tag{6}$$

and the floe perimeter per ice area, a measure of sea ice fragmentation,

$$\mathcal{P} \equiv \frac{\int\limits_0^\infty r P(r)dr}{\int\limits_0^\infty r^2 P(r)dr} = \frac{\pi}{2}\frac{\langle D^1 \rangle}{\langle D^2 \rangle}. \tag{7}$$

These derived quantities are useful because they require no further information about the sea ice (such as its concentration) to compare against modeled FSDs. However, both $\bar{r}$ and $\mathcal{P}$ can represent only those floes whose size is larger than $r_{min} = D_{min}/2$, the smallest possible floe size sampled. For perfect power-law distributions beginning at a scale of $r_{min}$ or before, both metrics are functions of $r_{min}$. However, for the real FCDs measured here, a maximum floe size exists, and a power-law scaling is not found approaching $r_{min}$, so the use of such metrics is justified (see Sec. 4). Because of the finite sampling resolution of the altimeter, chords that would originate from floes with a diameter near the sampling resolution may not be observed, and thus $\langle D^n \rangle \leq A_n \langle r^n \rangle$. We explore this uncertainty in Appendix B. For a known floe size distribution, the error decreases exponentially as a function of the distributional moment being considered, though it can be large (20% or more) in pathological cases. For distributional tails characterized by observed scaling exponents (Stern et al., 2018b), and for moments considered here, this uncertainty can be determined systematically and vanishes for measurement spacings smaller than the radius of the most common floe size. This resolution error does not affect the analysis of the power-law hypothesis, as that analysis is focused on the distributional tail. However, because $\mathcal{P}$ is a proportional to a negative moment of the FCD, it is sensitive to changes in the number of small chord lengths. Because of the measurement uncertainty for smaller chord lengths we will focus instead on $\bar{r}$ which is a positive moment of the FCD.

## 2.1 Evaluating the Floe Size Power-law Hypothesis with Floe Chord Data

Suppose the FSD $P(r)$ has a power-law tail that begins at some specified value $r_1$. Then for $r > r_1$, $P(r) \equiv P(r; \alpha, C) = Cr^{-\alpha}$, for an unknown coefficient $C$ and power-law slope $\alpha$. Integrating Eq. 1 over all $r$,

$$S(D) = \int\limits_0^\infty \tilde{F}(D;r)P(r)dr, \tag{8}$$

where the integral of the left-hand side of Eq. 1 is equal to $S(D)$ as $\int F(r;D)dr = 1$. Under the assumption of Eq. 2, if $P$ is a power law, so is $S(D)$ (Appendix A). For circular floes,

$$S(D) = \frac{2C}{\pi}\int\limits_{r_1}^\infty \frac{r^{-\alpha}}{\sqrt{(2r)^2 - D^2}}dr. \tag{9}$$

Because of the sampling resolution of the altimeter there is a minimum resolved chord scale $D_{min}$. If $D_{min} \ll D^* \equiv 2 \cdot r_1$, there is an explicit solution for $S(D)$, a power-law distribution over the range $(D^*, \infty)$

$$S(D) = C \cdot B\left(\frac{1}{2}, \frac{\alpha}{2}\right) \frac{2^{\alpha-1}}{\pi} D^{-\alpha} \equiv C_\alpha D^{-\alpha}. \tag{10}$$

where $B$ is the beta function. The coefficient $C_\alpha$ is a multiplicative factor independent of size, and the power-law exponent for a FCD is the same as the exponent for FSD, where the two are related by Eq. 1.

Moments of a power-law tail can be evaluated explicitly (for $\alpha > n+1$),

$$\langle r^n \rangle = C \int_{r_1}^{\infty} r^{n-\alpha} dr = C \frac{r_1^{n+1-\alpha}}{n+1-\alpha}. \tag{11}$$

Then for both the FCD and FSD, the ratio of two moments is independent of the unknown coefficient $C$, i.e.,

$$R_{n,\epsilon} \equiv \frac{\langle D^{n+\epsilon-1} \rangle}{\langle D^{n-1} \rangle} = D_{min}^\epsilon \frac{n-\alpha}{n+\epsilon-\alpha}, \tag{12}$$

valid for $n + \epsilon < \alpha$. The power-law coefficient can be obtained for any $n$, $\epsilon$ as,

$$\alpha_{n,\epsilon} = n + \epsilon \frac{R_{n,\epsilon}}{R_{n,\epsilon} - D_{min}^\epsilon} = \text{ constant.} \tag{13}$$

In the analysis below we will arbitrarily select only $n = 0.5, \epsilon = 1$ for comparison (for scaling coefficients $\alpha > 1.5$, the bulk of reported power-law coefficients are in this range: Stern et al., 2018b). Because the observations will not be perfect power-law distributions, we will use $\alpha_{0.5,1} \equiv \alpha^*$ as an estimator. A second estimate of the power-law scaling coefficient, $\hat{\alpha}$, is computed via the maximum likelihood estimator (Muniruzzaman, 1957; Clauset et al., 2009; Virkar and Clauset, 2014) (details in Appendix C) as,

$$\hat{\alpha} = 1 + \frac{N}{\sum_{i=1}^{N} \ln \frac{D_i}{D_{min}}}. \tag{14}$$

where $N$ is the number of chords. If the power law hypothesis holds, then the two estimates of $\alpha$ agree, although the agreement of $\hat{\alpha}$ and $\alpha_{n,\epsilon}$ is not sufficient to confirm the power-law hypothesis. In the Supporting Information (Text S1 and File S1), we compare these two estimates when they are evaluated against synthetic datasets drawn from a true power-law distribution. The two agree even when the size of the data is relatively small ($N < 25$). While in practice Eq 13 is easy to apply, it only holds when $\alpha_{n,\epsilon} > n+1$, and unlike the method of Clauset et al. (2009), it does not allow for a robust statistical analysis of the power-law fit, and should only be used when the data is assumed to follow a power-law already.

## 3 Climatology and Trends in Floe Properties Derived from CryoSat-2 Altimetry

We apply the analytic technique described in Sec. 2 to a floe chord data set constructed from the CryoSat-2 radar altimeter processed by the Center for Polar Observation and Modelling (CPOM) over the period from October 2010-present (CPOM data

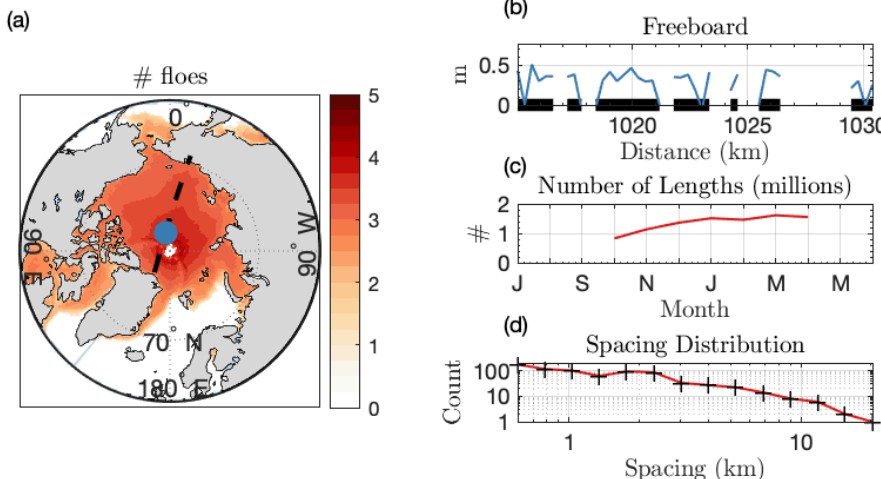

**Figure 2.** Constructing a FCD from altimetry. (a) Base 10 logarithm of the number of floe chords identified, binned into the CESM grid, across all CryoSat returns in the Arctic from 2010-2018. Black line is a single satellite track on January 21, 2014. (b) Subsection of the track centered on the blue dot in (a). Blue line is freeboard of sea ice in radar echoes defined as "floes" along the track. Black lines are chords identified from the freeboard retrieval. (c) Total number of chords measured in each month in the Arctic. Plot is centered on January 1. (d) FCD for the satellite track depicted in (a). Black marks on x-axis are the logarithmically spaced chord length bins.

products are available at http://www.cpom.ucl.ac.uk/csopr/seaice.html). CryoSat-2 radar echo returns are defined as "lead", "floe", "open ocean" or "ambiguous" according to waveform shape and sea ice concentration (Tilling et al., 2016, 2018b), at an approximately constant along-track spacing $D_{min} =$300 m. Floe chords are defined as a continuous sequence of one or more "floe echoes", with a gap of one ambiguous echo permitted within a floe sequence to allow for anomalous returns. A

5    chord length is taken from the midpoint of the first to the midpoint of the last radar echo. Individual chord lengths can be underestimated when continuous floes are separated artificially by producing two or more ambiguous echoes in sequence, or when highly reflective leads dominate the waveform return close to the floe edge and cause measurement dropout (Tilling et al., 2019). Lead contamination, or "snagging" (Armitage and Davidson, 2014) is more likely when the altimeter cuts of a small section of a floe, i.e. for small values of $\theta$. Overestimates of chord length can also occur when ice floes are in close contact

10    with neighboring floes. Therefore, floe chord lengths should be considered a satellite-derived product, not a true measurement of floe size. The minimum chord length retrieval $D_{min}$ is limited to the CryoSat-2 footprint ( 300 meters along-track) (see the discussion in Appendix B). However, surface discrimination via altimetry is highly accurate in months without melt ponds, (Peacock and Laxon, 2004; Guerreiro et al., 2017; Quartly et al., 2019), giving confidence that floe echos represent a coherent length of ice. More details on the details of chord identification may be found in Tilling et al. (2019). Indeed, this raw floe chord

15    data has been used successfully to reduce biases in altimeter-observed satellite sea ice thickness estimates from altimeters with different footprint sizes (Tilling et al., 2019). Here we analyze the sea ice floe size distribution using that floe chord product.

Figure 2 shows an example of floe chord data for a single CryoSat-2 track over the Arctic on January 21, 2014. Freeboard values for echoes discriminated as "floe" are plotted in Fig. 2b as a function of the along-track distance in km, and correspond to the blue circle in Fig. 2a. Floe chords are identified as black segments in Fig. 2b. The histogram of all 741 identified chords for this single satellite pass is shown in log-log space in Fig. 2d.

The full CryoSat-2 dataset examined here spans the time period from October 2010 to November 2018, and floe chords measured using the above technique are binned into the CICE sea ice model's two-dimensional sea ice grid for each month and year to facilitate comparison with model products. This implies that we invoke the principles of isotropy, homogeneity and stationarity of the FCD, required to produce such a distribution, on the length scale of the CICE model grid (O(25 km)) and time scale of a month. For every grid cell $i$, month $m$, and year $y$, we have a vector of floe chords $\{D_{i,m,y}\}$ from which we

build a FCD. The base 10 logarithm of the total number of floe chords recorded in each grid cell per month is shown in Fig. 2a. Because the satellite passes are densest near the pole, the measurement density is highest near the pole as well. Fig. 2c shows the number of Arctic measurements in each month. Sea ice type from CryoSat-2 is not available during summer months, as melt ponds make it difficult to discriminate between leads and ponded floe surfaces, and we do not include measurements from May to September. Across the entire set of satellite tracks included here, 11 million chord lengths are recorded in the Arctic.

Figure 3a shows the seasonal cycle of Arctic representative radius over the CryoSat-2 period obtained by applying Eq. 6 to the binned CryoSat-2 floe chord product. Individual years are plotted as thin lines, and the climatological average is shown in red. Details on how temporal and spatial average statistics are computed is included in Appendix D. During the months of October-December, the climatological representative radius is roughly 35% larger (7.06 km vs 5.18 km) than February-April. This seasonal cycle is broadly consistent across years. A possible interpretation of this seasonal cycle is that large first-year

ice pans form in October and are later fractured into smaller floes throughout the winter months. This concept is supported by observations that large-scale fracturing of sea ice in the Beaufort Sea is dominated by coastal processes and therefore only can occur once sea ice freezes to the coast in mid-winter (Richter-Menge, 2002), although such an interpretation is speculative and must be evaluated further as this method is refined. Fig. 3b shows annual-average representative radius in red for each full year from 2011-2017, with thin lines corresponding to the individual months within that year. Seasonal variability is significantly

larger than inter-annual variability. There is no statistically significant linear trend at the p=0.05 level.

The geographic variability of representative radius over the "early winter" (October-December) and "late winter" (February-April) periods are shown in Fig. 3c-d, for all grid areas. We display only those areas with at least 25 recorded floe lengths in each month during the averaging period. In Supporting Information Text S2 and Fig S1, we examine the sensitivity of bulk FSD statistics to this threshold, finding similar seasonal cycles and climatologies. The largest representative radii in the Arctic

lie in the interior Arctic near the pole, with a tongue of large floes that extends along the Canadian Arctic in late winter. There is a notable increase of representative radius with latitude. In the Supporting Info Fig. S2, we show that this relationship cannot be explained as a result of the increasing density of measurements near the pole and may therefore be a geophysical signal. The smallest representative radii (below 1 km) lie in Bering Strait and the Russian Arctic in early winter, and in the Laptev Sea in late winter. The difference in representative radius between fall and spring is accounted for by the reduction of floe sizes in

regions near the Arctic interior (see Fig. 6).

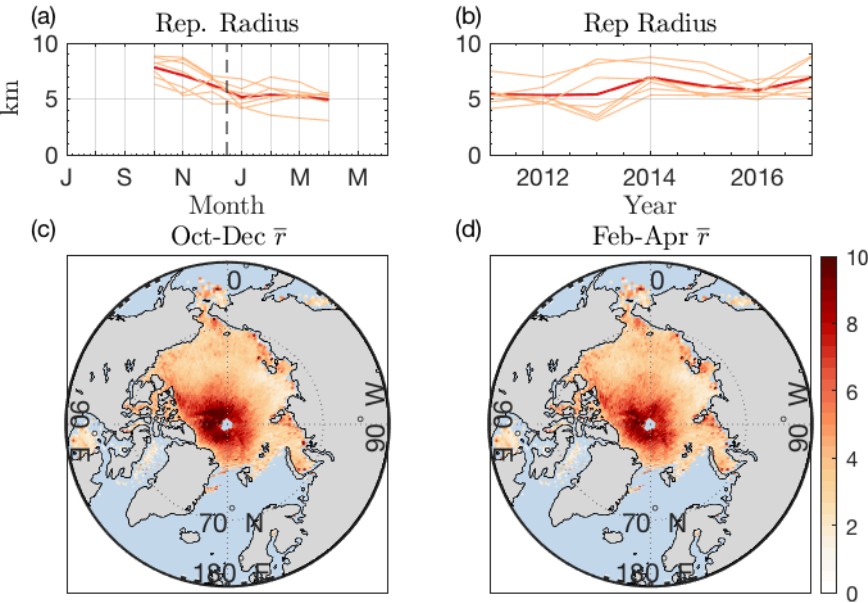

**Figure 3.** Top row: Temporal and geographic variability of Arctic representative radius. (a) Climatology of Arctic-average representative radius in units of km (red line). Thin lines are individual CryoSat-2 years. (b) Annual-average Arctic representative radius (red line). Thin lines are average in individual months. (c) Climatological representative radius in months October-December. (d) Same as (c) but for February-April.

## 4  Evaluating the Power-law Hypothesis Using Floe Size Statistics Derived from CryoSat-2

Given a collection of chord lengths, we would like to examine whether it is distributed according to a power law. Under the assumptions of Sec. 2, the scaling behavior of the FSD is the same as of the FCD (see Appendix A). We use the statistical methodology outlined in (Clauset et al., 2007, 2009; Virkar and Clauset, 2014) (which we term the MLE method) to evaluate shape parameters of the most likely power law fit and to test its plausibility. This method has been used to evaluate power-law behavior in recent FSD model (Horvat and Tziperman, 2017) and observational studies (Hwang et al., 2017; Stern et al., 2018b) and proceeds as follows:

1. **Lower-truncate the FCD**. First identify a minimum chord scale, $D^*$, above which we hypothesize a power law tail, and analyze only those floe chord measurements. We either (a) choose $D^*$ to be 900 m (to reduce the impact of small-size sampling errors discussed in Sec. 2) or (b) use the scheme described in Clauset et al. (2007) to evaluate the most likely value of $D^*$ for a power law tail. The length of this lower-truncated distribution is $N$. In the descriptions that follow, we use the subscript $all$ to describe case (a) and $tail$ to describe case (b).

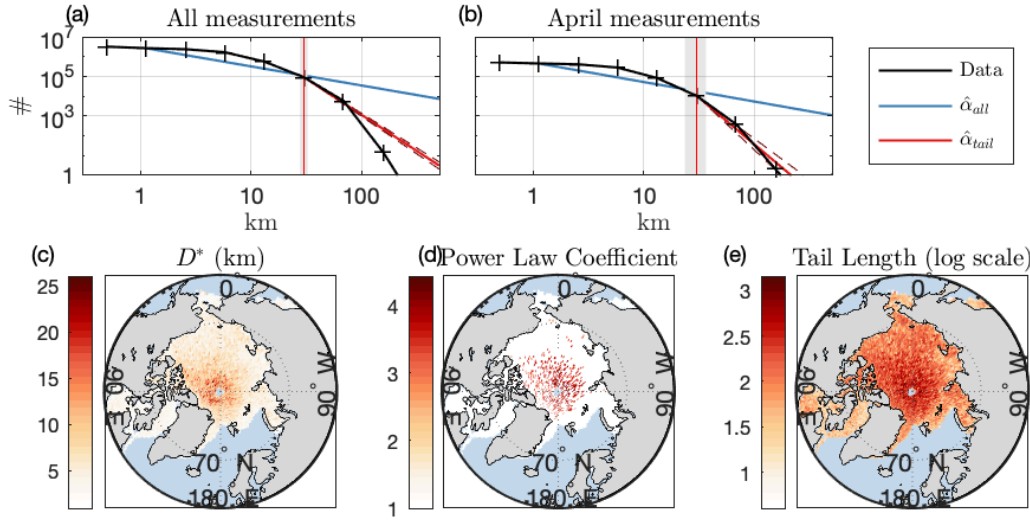

**Figure 4.** Examining the power-law hypothesis. (a) Histogram of all chord lengths recorded in the Arctic for the months November-April (black). Bin centers indicated by hashes and are logarithmically spaced. Blue line is power-law fit to all observed sizes according to eq. 14. Red line is power-law fit to the tail. Dashed red lines are fit lines using the $\pm 1$ standard deviation values of $\hat{\alpha}$. Red vertical line is the most likely beginning of the power law tail, $D^*$, with shaded region $\pm 1$ standard deviation in $D^*$. (b) Same as (a), but for measurements in April. (c) Maximum likelihood estimate of the beginning of the power law tail, $D^*$ (in km) for all measurements at each geographic location over the observational period. Only locations with $N > 1000$ are plotted. (d) Maximum likelihood estimate of power law tail exponent, $\hat{\alpha}_{tail}$, for the same points. Colored values have more than 200 chord lengths in the tail and $p > 0.1$. Zero values are those locations plotted in (c) but where either $p < 0.1$ or there are less than 200 measurements in the tail. (e) Number of chord lengths in the tail (above $D^*$) at each location.

2. **Compute power-law scaling estimates and parameter uncertainty**. We obtain two estimates of the FCD scaling estimate: either computing $\alpha^*$ via Eq. 13, or computing $\hat{\alpha}$, and uncertainty estimates in both $\hat{\alpha}$ and $D^*$ via the MLE method (Eq. 14). That the two estimates of $\alpha$ agree is a necessary condition for the FCD (and thus FSD) to be power-law distributed.

3. **Examine the plausibility of the power law fit**. We generate $M$ FCDs of size $N$ (the same number of synthetic chords as observed chords), with each synthetic FCD drawn from the hypothesized power law distribution $P(\hat{\alpha}, D^*)$. For each of these synthetic FCDs, we compute the Kolmogorov-Smirnov distance between it and the hypothesized power law model that generated it, $P(\hat{\alpha}, D^*)$. We also compute the distance between the observed FCD and $P(\hat{\alpha}, D^*)$. A p-value, $p$, is equal to the fraction of those $M$ synthetic FCDs that are "further away" from the hypothesized power law model than is the observed FCD. We use $M = 10,000$, which permits computation of $p$ within 0.005 (Clauset et al., 2009), and rule out the power-law hypothesis under the condition $p < 0.1$ (Virkar and Clauset, 2014).

We note that a "power law" describes the scaling of a distribution's tail. Previous observational studies have discussed "double power laws" (i.e., Toyota et al., 2011), i.e. two power-law distributions of different exponent joined at a specified scale. The methods employed here would capably capture the large-size power law scaling but not the small-scale scaling. Such "double power laws" are necessarily scale-variant, and require at least 3 parameters to describe. The conceptual and mathematical simplicity of the "power law hypothesis" does not apply in such a case, and we do not consider them here.

The MLE method is a rigorous test of the power-law hypothesis that eliminates potential human bias when interpreting observational data. To illustrate why this is important, we first consider the entire set of 11 million chord lengths recorded in the Arctic in all months (October-April), spanning a length range from 300 m to 100 km. The histogram of these floe chords is the black line in Fig. 4a (hashes on black line are the logarithmically spaced bin centers). Beginning from $D^* = D_{min} = 900$m, $\hat{\alpha}_{all} = 1.97$ (blue line) and $\alpha^*_{all} = 2.05$ (not shown). The observations are further away from synthetic data drawn from $P(\hat{\alpha}_{all}, D^*)$ in each of the $M = 1,000$ random draws ($p_{all} = 0/1000$) and we reject the power law hypothesis for these measurements. We note that if the resolution bias explored in Appendix B proves to be larger than expected, the under-representation of small floe lengths may affect the analysis of the full distribution.

Examining the tail of the distribution in Fig. 4a, the maximum likelihood estimate of $D^*$ is $\approx 15.0$ km (red vertical line, vertical shaded region is the range of uncertainty for $D^*$), above which there are $\sim$40,000 chord length measurements between 24.7 km and 99 km (0.4% of the dataset). On the truncated FCD, $\hat{\alpha}_{tail} = 4.65$ (red line, dashed lines are uncertainty ranges for $\hat{\alpha}_{tail}$), and $\alpha^*_{tail} = 4.67$ (not shown), similar to the large-scale roll-off reported in observations (Toyota et al., 2016). Even when restricted to the FCD tail, $p_{tail} = 0/1000$.

Finding no statistical basis for a power-law fit to the tail in Fig. 4a underscores the challenge in using the human eye to observe power law scaling. While the black and red lines in Fig. 4a appear similar across much of the range of sizes above 24.7 km, examining the misfit between the power law estimates and the data show that the two curves in fact differ significantly across the entire fit range. A misfit error can be defined as,

$$E = \left\langle \frac{|P(x_i, \hat{\alpha}_{tail}, D^*) - P(x_i)|}{P(x_i, \hat{\alpha}_{tail}, D^*)} \right\rangle \tag{15}$$

where the $x_i$ are the bin locations, angle brackets denote an average over the relevant bins, and $P(x_i)$ are the observed histogram values. Over the range from 24.7 km to 100 km, the misfit error is 33%. The visual agreement, misfit error, and apparent slope and shape of the distribution depend sensitively on the bin spacing and the logarithmic plotting.

Sea ice parameterizations that assume a power law distribution may significantly bias sea ice statistics. The imposition of any fixed distributional shape, when FSD dynamics are scale-variant, leads to implicit non-local redistribution of sea ice between floe size categories (Horvat and Tziperman, 2017). To see this in practice we compare the difference in Arctic-wide representative radius, $\bar{r}$, which is used in parameterizations of wave attenuation and ice thermodynamics, between the most-likely power-law fit to the data and the "true" value obtained via Eq. 6. The observations yield $\bar{r} = 10.2$ km, versus 34.5 km for the power-law fit. Examining only the tail of the distribution (chord lengths above 24.7 km) yields better agreement: 23.7 km for the observations and 24.4 for the fit line. Yet this tail constitutes just 1% of all measured chord lengths, corresponding to just 18% of total ice area and 4.5% of the perimeter per square meter (Eq. 7).

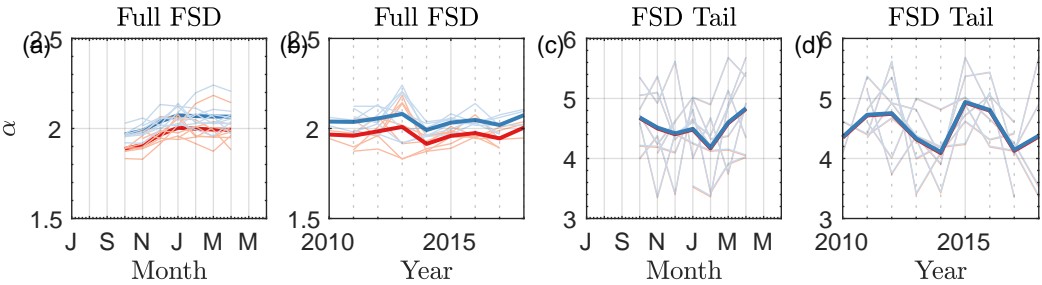

**Figure 5.** Top row: Temporal variability of power law fits to Arctic FCDs. (a) Estimate of the most likely power-law scaling coefficient for all recorded floe chords as a function of month over all years, calculated from the MLE method Eq. 14 (red lines) or Eq. 13 (blue lines). Thick lines are climatological averages, thin lines are individual years. Plot is centered on January 1. (b) Like (a), but plotted for individual years over all months. Thick lines are average over months plotted in (b) and thin lines individual months in each year. (c-d) same as (a-b), but for the distributional tail starting from $D^*$ computed using the MLE method. "Arctic" refers to points above $60°$N

Segmenting the chord length data into individual months in the Arctic, there are none where $p_{all} > 0$. Examining only the tail of each month's distribution, $p_{tail} < 0.1$ in all months. Only in April is there a non-zero $p_{tail} = 0.04$, for which the analysis of Fig. 4a is repeated as Fig. 4b. In April, $\hat{\alpha}_{all} = 1.99$,, $\hat{\alpha}_{tail} = 5.70$, and $D^* = 30.7$ km. The tail consists of 1618 measured chord lengths up to 97.5 km, accounting for 8% of the total floe area and 1.4% of the perimeter per square meter. The misfit
error between the April FCD tail and $P(\hat{\alpha}_{tail}, D^*)$ is 76%. Accumulating all measured chord lengths from October-May into the CESM model grid, we find zero locations that support a power law distribution across the range of measurements (i.e., $p_{all} > 0.1$). For grid areas with $N > 1000$, we show the value of $D^*$ computed using the local FCD in Fig. 4c. Values of $D^*$ range from 2 kilometers along the Russian Arctic to more than 10 km near the Pole.

While most of the Arctic has at least 1000 total measurements across all years, FCD tails ($D > D^*$) are not as well-sampled.
We investigate these tails including regions with at least 200 measured floe chords larger than $D^*$. The percentage of geographic areas with at least 1000 total measurements that have a tail with at least 200 measurements is 44%; on average $D^*$ is 5.4 km for these regions. For most of these regions we can not rule out a power-law tail. For the subset of regions with 1000 total measurements, 200 measurements in the tail, and where the power law hypothesis cannot be ruled out, the average $D^*$ is 6.5 km and average $\hat{\alpha}_{tail}$ is 3.34, within the typical range of Arctic FSD measurements (Stern et al., 2018b). In fig. 4d we show the
values of $\hat{\alpha}_{tail}$ at these locations. Colored cells are those with $p > 0.1$ and a tail with at least 200 measurements. In Fig. 4e we show the base 10 logarithm of the MLE tail for all geographic locations. Those regions for which a power law cannot be ruled out are generally those with the largest floes and the highest sampling, clustered near the central Arctic. The weakest support for a power-law tail is in the Chukchi and Beaufort seas, where power-law floe size distributions have often been reported. We note that our choice of tail length plays an important role in whether the power-law hypothesis is rejected in the tail across the
Arctic. For example, the fraction of Arctic regions with at least 1000 total measurements, a tail of at least 100, 200, and 400 measurements, and that does not reject the power-law hypothesis is 72%, 52%, and 15%, respectively. The better-sampled the FCD/FSD, the more likely the power-law hypothesis is rejected.

Scaling coefficients can provide useful information about the distributional shape. In Fig. 5(a-d) we show the seasonal and inter-annual variability of power-law estimates in the Arctic. Figure 5a plots the climatology of the power law scaling estimates when including all measured chord lengths in dark red (using Eq. 13) or blue (using 14). Individual years are thin red or blue lines. The two estimates disagree. Because agreement between the two estimates is necessary for the power law hypothesis to be true (see Sec. 4, SI Text S1), this alone is sufficient to rule it out. There is a seasonal cycle in the power-law fitting to the full distribution, with $\alpha_{all}$ increasing (steepening) from September to January and remaining flat until April, and no significant linear trend at the p=0.05 level for the annual-average value of $\alpha_{all}$. Fig. 5(c-d) repeats this analysis on the tail of those monthly distributions. In this case, the two estimates agree well. There is a different seasonal cycle in the steepness of the distributional tail: shallowest in early winter and steeper in late winter. This indicates that the changes across the winter months may be due to a reduction of the largest floes and a steepening of the distributional tail, although there is significant inter-annual variability among these estimates. A similar seasonal cycle to that found in Fig. 6(a,c), with an FSD that steepens from September to April, was found in image analysis of floes in the Beaufort and Chukchi Seas (Stern et al., 2018b), with $\alpha \approx 2.5$, although the distribution steepened monotonically over that period. There is no significant linear trend at the p=0.05 level in the annual-averaged FSD tail slope (Fig. 5d).

## 5 An Example Model-observation Comparison of Floe Size Variability

A principal aim of this work is to allow model-data comparisons and facilitate testing rapidly-developing FSD/FSTD models. Here we demonstrate how such a comparison can be made and provide useful information to modelers, even in the presence of the high uncertainties in this nascent FSD reconstruction technique. With the gridded data provided above, we may now directly compare development-stage sea ice models that incorporate FSD effects to observations. To do so, we use the Roach et al. (2018a) prognostic model for the FSD/FSTD, based on the Horvat and Tziperman (2015) theoretical FSTD framework, implemented into CICE 5.1.2 (Hunke et al., 2015) sea ice model. The FSTD is a sea ice state variable, subject to interaction of five key physical processes: lateral growth, lateral melt, fracture by ocean surface waves, welding of floes in freezing conditions and wave-dependent new ice growth (Horvat and Tziperman, 2015, 2017; Roach et al., 2018a, b). Previously published model runs (Roach et al., 2018a) focused on the impact of the FSD on lateral melt, which is largely driven by small floes (Steele, 1992), and so floe sizes above 1 km were not considered. As a larger range of scales is resolved in the CryoSat-2 observational product, we conducted a model run that extended the floe size categories to scales larger than 1 km, using 24 logarithmically-spaced floe size categories from 0.5 m to 33 km.

This FSTD model simulation is coupled to a slab ocean model and the WAVEWATCH III ocean surface wave model (Tolman, 2009), forced by the JRA55 atmospheric reanalysis (JRA-55, 2013) over the period from 2000-2016. These wave-coupled runs are branched at year 2000 from a standalone sea ice run from 1975-2000, spun-up using repeated 1975 atmospheric forcing. Additional model physics beyond those processes outlined in Roach et al. (2018a), have been added to determine the initial size of newly formed sea ice floes as a function of the ocean surface wave field. Details on this new parameterization, model initialization, and spin-up, are described in Roach et al. (2019). Recalling the finite measurement resolution of the CryoSat-2

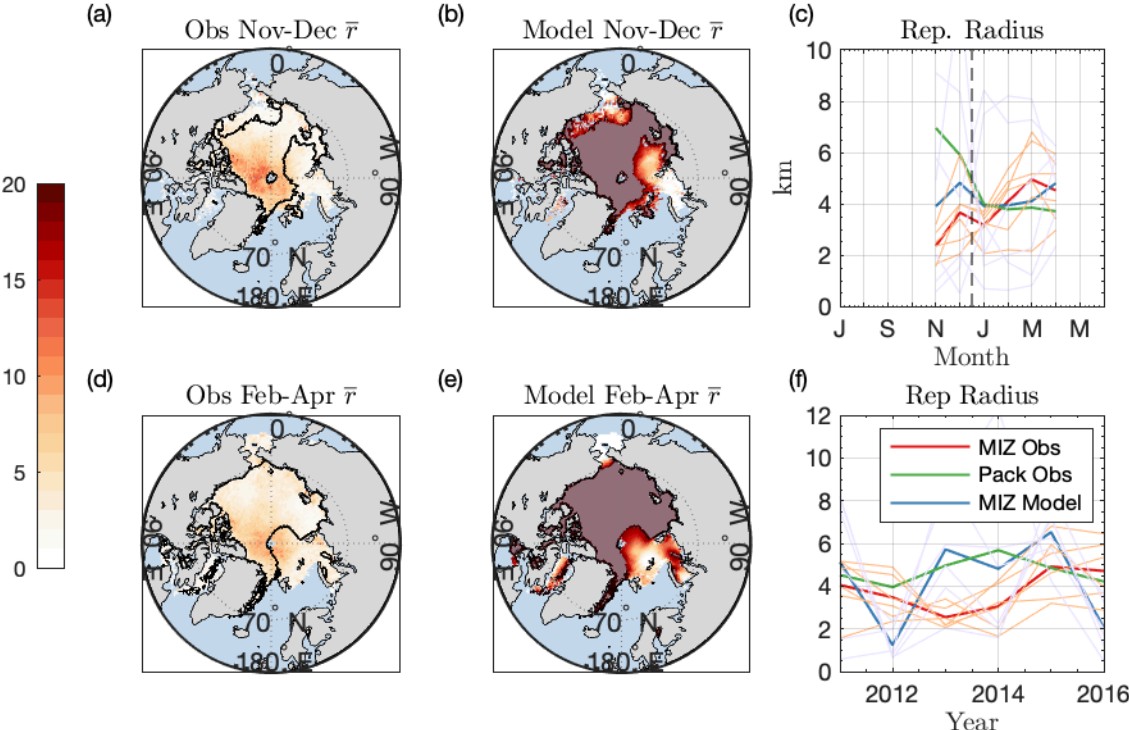

**Figure 6.** Geographic and climatological comparison of modeled and observed representative radii. (a-b) Average representative radius from November-December in (a) the CryoSat-2 observational dataset and (b) the FSTD model. Grey shaded regions in (b) are the interior of contours in (a), which represent "pack ice" unaffected by waves in the model simulations. (c) Climatology of Arctic-average representative radius in units of km for the MIZ in observations (red) and modeled (blue). Green line is the annual average for the "pack", the excluded regions in (b). Thin lines are averages in individual years from 2011-2016 in the MIZ. (d-e) same as (a-b), but for the months of February-April. (f) Annual-average Arctic representative radius for wave-affected regions in MIZ observations (red), MIZ model (blue), and pack ice observations. Thin lines are average in individual months in the MIZ observations.

dataset, the modeled representative radius is calculated only including floe size categories from 150 m (as floe sizes are radii, this corresponds to a radius equal to the minimum chord length) and larger. We include all FCD measurements here (chord lengths above 300 m) to make the broadest comparison, but note that the potential under-representation of floes with diameters near the sampling resolution may lead to inaccurate values of $\bar{r}$ in regions mainly consisting of such floes.

Fig. 6(a-b,d-e) compares modeled and observed climatologies of Arctic representative radius (for floe diameters 300 m and larger) averaged over 2011-2016 and the months of October-December (a,b) and February-May (c,d). Geographic variability of representative radius is broadly similar between model and observation: the largest floes lie in the Arctic interior, with regions of smaller floes in the straits and continental margins. Across the interior Arctic, simulated representative radii are significantly larger than are found in the observations, as the Roach et al. (2018a) FSTD model does not include processes that break up

large floes in the absence of ocean surface waves. To compare seasonality between model and observations, we compare only those regions that experience wave fracture in the model runs, areas we collectively term the marginal ice zone (MIZ). The MIZ is defined by excluding categories that do not experience wave fracture in a given month (see Appendix D), shown as the contoured regions in Fig. 6(a-b,d-f) and greyed out in Fig. 6(b,e)). All excluded "pack ice" regions have modeled representative radii greater than 18 km. The MIZ region accounts for 37% of grid areas with at least 25 chord measurements in months from

October-December and 35% of such areas for the period February-March. Note that the month of October is absent from these plots, because no well-sampled regions are classified as MIZ across all model years according to the criteria outlined in Appendix D.

Fig. 6(c) compares the observed (red) and modeled (blue) Arctic-average representative radii for the MIZ across over the period 2011-2016 as in Fig. 3(a). The seasonal cycle of representative radius in the MIZ is different in the observations (red

line, thin orange lines are individual months) than when all geographic regions are included (Fig. 3a). The seasonal cycle of representative radius in the "pack ice" region (i.e. not the MIZ) is shown as a green line in Fig. 6c. In the MIZ, average representative radii are smaller (on average 4.17 km vs vs. 6.49 km in the pack ice region). In contrast to the seasonal variation across all geographic regions (Fig. 3a) as well as in the pack ice, floes are larger in February-April than in November-December (5.40 km vs 3.15 km). In both the MIZ and pack ice regions, however, average representative radius is similar in late winter.

The largest difference between the two regions is from November-December, where representative radii are more than twice as large in the pack ice than the MIZ.

Fig. 6(f) shows the annual average representative radius in the MIZ (red), pack ice (green) and modeled MIZ regions (blue). Modeled MIZ representative radii have a similar magnitude compared to the MIZ observations, though these regions have smaller floes than the interior. To address the scale mismatch between the too-high modeled floe sizes and observed represen-

tative radii in the interior Arctic, as well as the strong and different seasonal cycle in representative radius in both regions, modeling efforts must include additional mechanisms for reducing floe size in the Arctic interior away from waves, such as mechanical fragmentation (Toyota et al., 2006; Rynders et al., 2016) or ridge dynamics (Roberts et al., 2019), to obtain realistic representative radii across the entire Arctic, as these processes are not present in the model used to make this comparison.

## 6 Conclusions

Here we developed and demonstrated a method for deriving the statistics of the sea ice FSD from satellite radar altimeter measurements of chord length. This method provides the first pan-Arctic accounting of climate-relevant quantities derived from the FSD, permits testing of existing scaling laws previously used to characterize distributions of floe size, and allows for
gridded comparisons between FSD models and observations. Using this new technique we produced climatological, annual-average, and geographic mean moments of the Arctic FSD across a range of resolved length scales from 300 m to 100 km.

With the combination of satellite altimetry and mathematical theory, we were able to rigorously examine the "power-law hypothesis" related to the FSD under simple assumptions about the underlying floe chord data and the fidelity of CryoSat-2 satellite retrievals. Segmenting measurements by geographic location, by month, and by year, we find limited statistical
basis for a power-law scaling beginning below about 6.5 kilometers. In a limited number of geographic locations, we find the observational data cannot rule out power-law scaling, but for typical sizes above about 6.5 kilometers. Assuming a power-law floe size distribution can bias sea ice model output and conceptual understanding. The geographic variability and lack of consistent multi-scale behavior reinforces the need for sea ice models to account for floe-scale processes rather than diagnose a distributional shape.

Observations that span the polar regions and different years and seasons are valuable for future refinement of process-based models of the FSD. In Sec. 5, we demonstrated how such model-observation comparisons can be made and can provide useful insights for model developers. At present, some general features of floe size evolution (in particular the magnitude and seasonal cycle of the representative radius) are broadly similar between model and observation in the marginal ice zone. Yet there is a significant scale mismatch in the interior Arctic between the presented simulations and this observational product, because of
missing fragmentation physics in the absence of ocean surface waves. Floe size modeling efforts have focused on marginal ice zone processes (Horvat and Tziperman, 2015; Zhang et al., 2015), and particularly floe sizes below about 1 km because these small floes play an important role in sea ice thermodynamics for floe sizes. The CryoSat-2 observations, however, are best suited to resolving floe chords of several hundred meters and above. New satellite altimeters like ICESat-2 have the potential to increase the chord length resolution to scales of 20-100 m and provide insight at smaller scales.

We emphasize strongly that refinement may be necessary to apply this method for operational purposes, trend analysis, and further model validation. This paper has focused on the framework for making altimetric measurements of the FSD and comparison to model output, but the obtained chord lengths and distributions have not been carefully validated against other observational methods, and this will be necessary before further application of this method. Before doing so, we have tried to outline the most significant uncertainties in the method. The typical assumptions of homogeneity, isotropy, and stationarity are
invoked here at the length scale of the CICE model grid (O(25) km on each side) and time scale of one month. These statistical assumptions may not be satisfied if, for example, the number of measurements in a given region in one month is insufficient to sample the known anisotropy of the sea ice floe field, and additional passes change the mean chord length significantly (see Supporting Information Text S2 and Fig S1). While we found little evidence for power-law scaling throughout most areas of the Arctic, this may be sensitive to the geographic (here the CICE model grid of approximately 25 km x 25 km) and temporal

(here all measurements from 2010-2018) windows we use to collect and evaluate chord length measurements for a power law. The assumption of scale-invariant sampling, observational uncertainty because of the finite sampling resolution, analysis of ambiguous returns, and the accuracy of retrievals in regions of thin sea ice may also affect the inferred size of sea ice floes. This in turn may affect the climatologies described in this study.

While processed CryoSat-2 data has been validated against both visual imagery and ground-based observations, it was not designed with this application in mind — additional quality control may be necessary for climate studies of changing floe properties. The positive comparison between model and observation in Section 5 could also be due to a compensation between these measurement uncertainties and will need to be re-examined in future validation work. Yet observational uncertainties regarding, for example, the floe shape distribution can be roughly estimated at the order of the error in "effective radius"

obtained for circular floes ($r = \sqrt{A/\pi}$) or a square ($r = \sqrt{A/4}$), a relative error of 25%. To constrain model results beyond this scale of error will require further refinement. However, as shown in Fig. 6, at present the model-data mismatch in the interior Arctic can exceed a factor of 3. Even with expected levels of error in the present derived FCD/FSD product, some constraints on the model can be considered at present with this method. A future comparison of results from the the Ice-Sat2 and CryoSat-2 altimeters will provide insights into the relevance of measurement and statistical uncertainties, as will

comparison of altimetrically derived floe chords measurements with visual imagery.

    Even accounting for important caveats that arise from making satellite measurements, remotely sensing the sea ice FSD from altimeters at sub-daily resolutions can provide a significant increase in data for comparison and analysis of new sea ice models that parameterize the FSD. Previously the difficulty of making measurements of the FSD at relevant spatial and temporal scales has inhibited the wide-spread adoption of such floe-sensitive sea ice models. Understanding sea ice variability at the floe scale

is also an important aspect of sea ice forcasting, and the ability to remotely assess the sea ice FSD at near-real-time will allow for further improvement of operational forecasting networks.

*Data availability.* CPOM sea ice data, including raw floe length data, are available on the CPOM data portal at http://www.cpom.ucl.ac.uk/csopr/seaice.htm The processed FCD/FSD statistics are available at https://github.com/chhorvat/CRYOSAT-FLOES/. The Roach et al. (2018a) FSTD model is publicly developed and available at https://github.com/lettie-roach/.

**Appendix A: Proof that the FCD and FSD have the same statistical properties**

For generic probability distributions $S(D)$ and $P(r)$, and a probability function $\tilde{F}(D;r)$, via equation 4 we have the relationship,

$$\langle D^n \rangle = \int\limits_0^\infty dr\, P(r) \int\limits_0^r D^n \tilde{F}(D;r)\, dD. \tag{A1}$$

Where we restrict the upper bounds on the second integral because $\tilde{F}(D;r)$ is zero for $D > r$. Under the scale-invariant sampling assumption $\tilde{F}(D;r)dD = G(\xi)d\xi$, where $\xi = \frac{D}{2r}$ for $D < 2r$ ($\xi < 1$). Therefore,

$$\langle D^n \rangle = \int_0^\infty dr\, P(r) \int_0^1 r^n \xi^n G(\xi)\, d\xi \tag{A2}$$

$$= \int_0^\infty dr\, P(r) r^n \int_0^1 \xi^n G(\xi) d\xi \tag{A3}$$

$$= A_n \cdot \langle r^n \rangle, \tag{A4}$$

where $A_n$ is the $n^{\text{th}}$ moment of $G(\xi)$, a constant that depends on the functional form of $G$. For any such probability function (for example that derived in Sec. 2 for circular floes), the moments of the FSD and the moments of the FCD are proportional. Most of the hypothetical statistical distributions we would consider (for example, power laws) can be fully determined in terms of their moments, and thus the relationship between moments of the FSD and FCD is typically sufficient to reconstruct the underlying FSD.

Supposing $P(r)$ was a power-law function, converting Eq. 8 to an integral over $\xi$ from 0 to 1, we have,

$$S(D) = \int_0^\infty \tilde{F}(D;r) P(r) dr = \int_0^1 \frac{P(D/2\xi) G(\xi)}{\xi} d\xi. \tag{A5}$$

For a power-law function, $P(D/(2\xi)) \propto \left(\frac{D}{\xi}\right)^{-\alpha}$ and

$$S(D) \propto \cdot D^{-\alpha} \int_0^1 \xi^{\alpha-1} G(\xi) d\xi = A_{\alpha-1} D^{-\alpha}. \tag{A6}$$

From Equations A4 and A6, and under the assumptions of Sec. 2, all moments of the FSD and FCD are related by a computable function of the moment only, and power-law FSDs are derived from power-law FCDs with the same scaling law. While the proportionality of moments and Eq. A6 prove that an observed power-law FCD must reflect an underlying power-law FSD, the same analysis used to arrive at Eq. A6 can be repeated to find $P(r)$ given a power-law distributed $S(D)$ as well.

### Appendix B: Bounds on the Relationship between Chord Length and Floe Size Moments

The real altimetric data product has a finite sampling resolution $D_{min}$ which can bias the computed FSD moments and power-law decay profile. For example, applied to real data with a finite sampling resolution , the integrals in equations 4 to 5 are taken beginning at the minimum observed chord lengths $D_{min}$ and floe sizes $r_{min} = D_{min}/2$. Moments of the distributions $S$ and $P$ reflect only statistics for floes larger than $D_{min}$ and $r_{min}$, respectively. All other aspects of this derivation remain the same,

as $\tilde{F}(D;r)$ is zero for any $r < D/2$. However, the relationship expressed in Eq. 4 becomes:

$$\langle D^n \rangle = \int_{r_{min}}^{\infty} dr\, P(r) \frac{2^{n+1}}{\pi} r^n \int_{Y(r)}^{\frac{\pi}{2}} \sin(x)^n dx \tag{B1}$$

$$= A_n \langle r^n \rangle \left[ 1 - \frac{\int_{r_{min}}^{\infty} dr\, P(r) \frac{2^{n+1}}{\pi} r^n S_n(Y(r))}{A_n \langle r^n \rangle} \right] \tag{B2}$$

$$\equiv A_n \langle r^n \rangle \left[ 1 - E(P(r);n) \right]. \tag{B3}$$

where $Y(r) \equiv \sin^{-1}(\frac{D_{min}}{2r})$, $S_n(y) = \int_0^y \sin^n(x)dx$, and $E$ is the error in relating the $n^{\text{th}}$ moments of $S(D)$ and $P(r)$. Since $P(r)$ is unknown, $E$ cannot be computed a priori. The function $S_n(Y(r))$ expresses the percentage of chords formed from floes of size $r$ that would be smaller than $D_{min}$, although it is not readily expressed as a function of $n$. The most pathological distributio is when $P(r)$ is a delta function at $r_{min}$, $P(r) = \delta(r - r_{min})$, $Y(r_{min}) = \pi/2$ and $E = 1$ as no chord lengths would be measured.

We can compute the error function for any delta function distribution as,

$$E(\delta(r - r^*);n) = \frac{S_n(Y(r^*))}{S_n(\frac{\pi}{2})}, \tag{B4}$$

and the misfit is the proportion of the integral of $\sin^n(x)$ betweeen 0 and $Y(r^*)$. Because $\sin(x)$ is monotonically increasing from $x = 0$ to $\pi/2$, the integral of $S_n$ is bounded above:

$$S_n(Y(r^*)) \le Y(r^*) \sin^n(Y(r^*)) = Y(r^*) \left( \frac{D_{min}}{2r^*} \right)^n, \tag{B5}$$

and the misfit error is bounded above by,

$$E(\delta(r - r^*;n)) \le \left( \frac{D_{min}}{2r^*} \right)^n \frac{Y(r^*)}{B(\frac{n+1}{2}, \frac{1}{2})}. \tag{B6}$$

The reciprocal of $B$ is equal to $\pi$ at $n = 0$ and decreases sub-linearly, and so away from $r_{min}$ the error term decays exponentially with $n$ and is small even for nearly-pathological distributions (for n=1, $r^* = D_{min}$, for example, $E \le \pi/24 \approx 14\%$. Knowing the distribution of errors behaves in this way allows us to establish upper bounds by integrating $P$ as a sum of $\delta$ functions.

We note that increasing resolution of floe chords will result in tighter bounds on this error. When $Y(r)^* \le 1$, which occurs when $r^* \ge \frac{D_{min}}{2\sin(1)} \approx 0.59 D_{min}$, we can exploit a tighter bound using the fact that $\sin^n(x) \le x^n$,

$$S_n(Y(r^*)) \le \frac{Y(r^*)^{n+1}}{n+1} \le Y(r^*) \left( \frac{D_{min}}{2r^*} \right)^n. \tag{B7}$$

Using the same example as above ($n = 1$, $r^* = D_{min}$) bounds the error $E \le \pi^2/144 \approx 7\%$. A real-world distribution of floe sizes must have a peak value above zero, thus by increasing the sampling resolution (say, for example, to near the size of

pancakes, i.e. $D_{min} \approx 20$ meters or less, approached by the ICESAT-2 altimeter), this bound takes over and errors are reduced substantially.

We can explicitly solve Eq. B3 for distributions with power-law tails. These distributions are peaked at the minimum floe size, and so will have high moment error. For power laws with $\alpha = -1, -2, -3$, or $-4$, $E(P(r; \alpha, r_{min}), 1)$ is 1, 4, 16, or 25 percent. For $n = 2$, $E(P(r; \alpha, r_{min}), 2)$ is .003, .04, 2, or 9.6 percent: the increase in error with decreasing $\alpha$ is because sharper power law slopes concentrate most of the distribution towards the smallest scale.

## Appendix C: Maximum Likelihood Estimation for Chord Length Distributions

Given a set of floe chords $\{D\}_i$ and an estimate of the beginning of a power-law tail $D^*$, we would like to find the most likely power-law floe size distribution $P(r; \alpha, r_{min})$ that generated them. As discussed in Appendix A, moments of the FSD and FCD are related by a multiplicative factor, and the distributions themselves will share the same power-law exponent. Thus we may test the power-law hypothesis directly on the FCD $S(D)$. The power-law hypothesis means that $S(D)$ is of the form,

$$S(D) = \frac{(\alpha - 1)}{D^*} \left( \frac{D}{D^*} \right)^{-\alpha}. \tag{C1}$$

Following (Muniruzzaman, 1957; Clauset et al., 2009) (see also the derivation in Stern et al. (2018a)), we compute the log-likelihood of the observations for a given $\alpha$ (eq. 10),

$$\mathcal{L} \equiv \ln \prod_{i=1}^{N} S(D_i) = \ln \left[ \left( \frac{\alpha - 1}{D^*} \right)^N \prod_{i=1}^{N} \left( \frac{D_i}{D^*} \right)^{-\alpha} \right] \tag{C2}$$

$$= N \ln(\alpha - 1) + N(\alpha - 1) \ln D^* - \alpha \sum_{i}^{N} \ln D_i. \tag{C3}$$

As the natural log is monotonically increasing in its argument, to find the most likely $\alpha$, denoted $\hat{\alpha}$, we take the derivative with respect to $\alpha$ and set to zero,

$$\frac{1}{\alpha - 1} + \ln(D^*) = \frac{1}{N} \sum_{i=1}^{N} \ln \frac{D_i}{D^*}, \tag{C4}$$

which resolves as a solution for the most likely $\alpha$:

$$\hat{\alpha} = 1 + \frac{N}{\sum_{i=1}^{N} \ln \frac{D_i}{D^*}}. \tag{C5}$$

The above analysis concerns the most likely $\alpha$ that explains the FCD. We may ask a separate question: what is the most-likely $\alpha$, which we define as $\alpha_P$, that would explain the FSD, given the explicit relationship that can be derived between $S(D)$ and a power-law distributed $P(r)$ examined in Eq. 10:

$$S(D) = C \cdot B \left( \frac{1}{2}, \frac{\alpha}{2} \right) \frac{2^{\alpha - 1}}{\pi} D^{-\alpha} \tag{C6}$$

where $C$ is unknown. Repeating the above analysis,

$$\mathcal{L} \equiv \ln \prod_{i=1}^{N} S(D_i) = \ln \left[ C^N B\left(\frac{1}{2}, \frac{\alpha_P}{2}\right)^N \left(\frac{2^{(\alpha_P - 1)}}{\pi}\right)^N \prod_{i=1}^{N} D_i^{-\alpha_P} \right] \tag{C7}$$

$$= N \ln C + N \ln B\left(\frac{1}{2}, \frac{\alpha_P}{2}\right) + N(\alpha_P - 1)\ln 2 - N \ln \pi - \alpha_P \sum_{i}^{N} \ln D_i. \tag{C8}$$

Next we take the derivative of $\mathcal{L}$ with respect to $\alpha_P$ and setting to zero. We use the fact that $B(x,y) = B(y,x)$, and $\frac{\partial B(x,y)}{\partial x} =$
$B(x,y)\left(\psi(x) - \psi(x+y)\right)$ where $\psi$ is the digamma function, to find,

$$\frac{\partial \ln B\left(\frac{1}{2}, \frac{\alpha_P}{2}\right)}{\partial \alpha_P} = \frac{1}{2}\left(\psi\left(\frac{\alpha_P}{2}\right) - \psi\left(\frac{\alpha_P + 1}{2}\right)\right). \tag{C9}$$

The maximum likelihood $\alpha_P$ is the solution to the transcendental equation,

$$\frac{1}{2}\left[\psi\left(\frac{\alpha_P}{2}\right) - \psi\left(\frac{\alpha_P + 1}{2}\right)\right] + \ln 2 = \frac{1}{N}\sum_{i=1}^{N} \ln D_i. \tag{C10}$$

which is an alternative method for obtaining the FSD scaling.

**Appendix D: Averaging and Segmenting FSD Statistics**

Due to limitations in the number of floe chords recorded at any particular location over time, we do not include all geographic locations when computing hemispheric means. Averaging is performed by including only geographic regions where there are at least 25 recorded floe chords. The area being averaged over is thus not fixed in time. For seasonal cycle plots, we only include months which have enough measurements for all fully-sampled CryoSat-2 years (2011-2018). For annual averages, we
include only those years where all CryoSat-2 months (excluding June-September) have enough measurements.

When masking additional regions to perform the model/observation comparisons in Fig. 6, we note that because the Roach et al. (2018a) model does not include processes that fragment larger floes into smaller floes in the absence of ocean surface waves, regions in the interior Arctic without wave activity have nearly all sea ice area belonging to the highest floe size categories. Nearly all regions where wave fracture is an active process also have representative radii below about 10 km (Roach
et al., 2019). We define regions that do not experience wave fracture as those with an abnormally high simulated representative radius, which we choose to be the 22[nd] floe size category ($\bar{r} = 18.6$ km) or above. The mask and comparisons in Fig. 6 are made by excluding all such areas.

*Author contributions.* CH derived the mathematical theory and wrote the manuscript. LR built and performed the climate model simulation. RT, AR, and AS provided and interpreted the CryoSat-2 data. KH, CG, CB, and BK contributed to the study design. All authors have
participated in manuscript preparation.

*Competing interests.* The authors declare no competing interests.

*Acknowledgements.* CH was supported by the NOAA Climate and Global Change Postdoctoral Fellowship Program, administered by UCAR's Cooperative Programs for the Advancement of Earth System Science (CPAESS), sponsored in part through cooperative agreement number NA16NWS4620043, Years 2017–2021, with the National Oceanic and Atmospheric Administration (NOAA), U.S. Department of
5 Commerce (DOC). CH, CG, and KH thank the American Mathematical Society for their support through the Math Research Community "Differential Equations, Probability, and Sea Ice", funded by NSF grants 1321794 and 1641020. LR was funded via Marsden contract VUW-1408 and the New Zealand Deep South National Science Challenge, MBIE contract number C01X1445. CMB was supported by the National Science Foundation grant PLR-1643431. BFK was supported by ONR grant N00014-17-1-2963 and NSF grant 1350795. RT, AR, and AS were supported by the UK NERC Centre for Polar Observation and Modelling and the European Space Agency

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
