# Peer review of "Estimating The Sea Ice Floe Size Distribution Using Satellite Altimetry: Theory, Climatology, and Model Comparison"

_The Cryosphere, 2019_

## Referee Comment (RC1) · Thomas Armitage (Referee) · 17 Jul 2019

**Review of "Estimating the sea ice floe size distribution using satellite altimetry: Theory, climatology, and model comparison", by Horvat et al.**

T. Armitage

The paper sets out a framework for estimating geophysical parameters relevant to the sea ice floe size distribution based on floe chord lengths measured by the CryoSat-2 radar altimeter. The paper exploits the fact that it is possible to distinguish between returns originating from leads and returns originating from floes, to estimate chord lengths as the distance over which consecutive CS2 waveforms are identified as floes uninterrupted by leads. These chord lengths are then related to the floe size distribution by statistical considerations. I was asked by the editor to assess the remote sensing aspect of this paper; I have not attempted to assess the theoretical developments because this is outside of my area of expertise, and I leave this to other reviewers. Based on the remote sensing aspect only, I would recommend publication after the authors address the following points.

P1 L8: "radio" should be "radar"

I would like to see a little more discussion on the limitations of the technique imposed by the sampling of the radar and its high sensitivity to areas of open water. Any amount of bright open water (lead) in the pulse limited footprint (300m along-track x 1.6km across-track) is likely to be identified as a lead or ambiguous. Also, bright open in the larger beam limited footprint (up to ~20km across-track) can potentially lead to an 'ambiguous' return (although this depends on the specific waveform processing). I think this could affect your processing in a few ways:

1. Your assumption that theta is uniformly sampled: The fact that the altimeter footprint isn't symmetric means that for large theta the footprint is almost tangent to the floe perimeter when they intersect, whereas for small theta the footprint is almost perpendicular to the floe perimeter where they intercept. This means that chords with small theta are more susceptible to 'snagging' or contamination by leads, particularly at the beginning and end of the chord closer to the floe edge. This means that the length of chords with small theta could be underestimated.
2. Your taking Dmin as ~300m (P8 L11): I think the minimum chord length (if you are taking a minimum of two consecutive floe measurements to represent the smallest possible chord) is closer to 600m. This is because for a waveform to be classified as a floe you really expect (at least) the pulse limited footprint to contain entirely sea ice and no open water – any amount of open water will dominate the sea ice return. The other side of this is then that you can only assign a range to Dmin (from around 600m up to maybe 1200m if the leads either side are at the forward and backward limit of the along track footprints) and within this range it is ambiguous.

Figure 3 and P9 L3-10: You claim that the largest representative radii lie along the Canadian Archipelago, but the maps show that the largest radii actually surround the 'pole hole'. It seems like there is a geophysical signal, with larger radii in the MYI zone, which I would expect, but then superimposed on that is a signal that seems to increase with latitude such that the largest floe sizes are close to the pole hole. I was wondering if you have an explanation for the large floe radii surrounding the pole hole, because this isn't something I would intuitively expect? The spatial sampling increases as you move north due to the convergence of orbits, could this skew the mean floe radius large?

Incidentally I couldn't find Figure S1?

---

## Short Comment (SC1) · 19 Jul 2019

As pointed out by Dr. Armitage, Supplementary Figure S1 was not included in the zipped archive. I have attached that figure here, and apologize for the mistake.

[Figure]

[Figure]

**Fig. 1.** Impact of chord number threshold on derived floe size statistics (see Text S2)

---

## Referee Comment (RC2) · Anonymous Referee #2 · 22 Jul 2019

General comments: This paper aims at developing a new method to obtaining the floe size distribution in the Arctic Ocean using the chord length data derived from the CryoSat-2 altimeter data. The conversion of the chord length distribution to floe size distribution statistically seems to the highlight of this paper. The authors accomplished this with a strong mathematical background and an assumption of circular shapes of ice floes and homogeneous and isotropic distribution. With this method, they attempted to show the geographical, seasonal, and interannual properties of the floe size distribution in the Arctic Ocean for the first time. They also tested the validity of power law distribution which has been applied for the floe size distribution frequently. As a result, they concluded that although power law scaling cannot be ruled out, the statistical basis is limited and especially the assumption of power law is weak in the Chukchi and Beaufort seas. They also emphasized the refinement of this method for the operational use.

Floe size distribution is one of the important parameters of sea ice and should be considered to understand the behavior of sea ice area. However, due to the limited observations, its statistical properties are not clarified yet. The idea of using the chord length is interesting in that it can cover wide areas and the truncation error caused by the traditional analysis of finite area seems to be reduced. Therefore, I agree this paper presented an interesting and useful method to analyze the floe size distribution. But to do so, I think the new method should be validated carefully with observational data and the consistency should be confirmed by comparing it with the traditional method. This paper focuses mainly on the availability of the new method with their strong mathematical background. While I feel this is an important step, the validation of this method seems relatively weak. In this case, I think the validation is especially important because their method is based on several assumptions such as circular shapes or homogeneous distribution. Thus, while I agree their method is interesting, my evaluation for this paper is somewhat reserved. In addition, the descriptions are not necessarily readable at places for me, and might be improved, I feel. The major points are as follows:

1) Assumptions of this method. The authors assume that the floe chord distribution data is homogeneous, isotropic, and stationary within the region, and time data is collected (P3L17-18). For simplicity in mathematical treatment, it might be allowed. But I think the validity of these assumptions should be examined somewhere in the paper. For example, fracture patterns caused by shear stress near the shore are far away from circular shapes (e.g., Schulson and Hibler, 1991 Journal of Glaciology). As a preliminary step, I encourage the authors to confirm this method is available even in such situation based on the real floe patterns. Besides, to make an assumption of stationary distribution, the time scale on which this assumption is valid should be discussed

[Figure]

because they discuss the seasonal and interannual variation of FSD later. If they examined that their method is applicable to obtain floe size distribution by comparing with the traditional analysis based on real satellite images, the value of this paper would have been enhanced significantly. Theoretical support with mathematics is important, but that is not enough, I think.

2) Technical matters * To represent floe size, they used a radius of the equivalent circle (P3L7). Is it common? To my knowledge, a diameter has usually been used in the past studies. I think a diameter fits the sense of floe size better, although this might be an essential matter.

* In the past studies, floe size distribution has been represented in two ways: cumulative number distribution and non-cumulative one. To avoid confusion, it might be better to declare which way this paper takes somewhere in section 2.

* In section 3, how they determined the individual chord lengths from the Cryosat-2 records is still unclear to me. When ice floes are contacting the neighboring floes, how did they determine the boundary? How much is the measurement accuracy of vertical distance? If it is about 10 cm, it would be quite difficult to identify the edge of the floe for thin ice especially. I think this is a critical matter.

* In section 4, they tested the validity of power-law distribution. It might be possible that the real FSD may have different regimes although the power-law is applicable for each regime. Figure 4 may suggest such possibility. In such a case, how do they judge the validity?

3) Interpretation of the results Overall, I feel the discussion of the results might be a bit weak compared with the preparatory statement about mathematical treatment. For example, they showed "During the months of October-December, the climatological representative radius is roughly 35% larger than February-April" (P8L32-34). I would like to know how they interpret this result because from intuition floe size tends to become larger at the later growth stage. I am wondering if the measurement accuracy

might affect this result significantly.

4) The applications of the results They examined the validity of power-law distribution (section 4) and the application to numerical sea ice models (section 5). But since the validity of this method is not fully investigated from the observations, I am wondering if these applications might be really useful. Especially, it was difficult for me to understand the authors' intention about section 5. Personally, at this stage I prefer focusing on showing the validity of this new method to extending the results to models.

Specific comments: *(P1L2) "spare" should be replaced by "sparse".

*(P1L15) "covered in sea ice" might be "covered with sea ice".

*(P1L18, P2L12) "Rothrock and Thorndike, 1984b" I could not see the difference between 1984a and 1984b in the reference lists. If they are the same, please take "b".

*(P2L19-20) This is true. But to describe this, it would be needed to show that FCD is better than FSD in accuracy.

*(P3L15) "the fraction of floe chords" should be "the number fraction of floe chords"?

*(P3L3) In Eq.1, "F(r;D) S(D)" can be replaced by "F(r;D) S(D) dD dr" to show the number of floes which have radius between r and r+dr. The same applies for the righthand term.

*(P3L24) I wonder if "F(r;D) dr dD" should be "F(r;D) S(D) dr dD".

*(P3L27) Likewise, I wonder if "F∼(D;r) dr dD" should be "F∼(D;r) P (r) dr dD".

*(P4L1) D/r should be from 0 to 2 (not 1).

*(P5L1) I wonder that since they consider a circle rather than a semi-circle, theta should take between 0 and 2*pi (not pi).

*(P5L1) Please state the definition of the function T.

*(P6L18) Please insert "under the condition, alpha > n+1" after "explicitly".

*(P6L23) In Eq.13, I wonder if "e" (epsilon) should be "1-e".

*(P7L5-12) Sorry, but I could not follow this paragraph. It would be helpful if you rewrite this paragraph with an emphasis of your intention.

*(P9L5-6) This sentence may contain grammatical error. Please rewrite it.

*(P9L9) "representative radius from fall and spring" might be "representative radius between fall and spring".

*(P11L11-13) I could not understand this sentence.

*(P12L4-5) "Assuming. . . parameterizations." is hard to follow.

*(P13L16) "The two. . . hypothesis." I could not understand why.

*(P13L29) "below 300 m" should be "below 30 m". Please check Steele's paper. Accordingly, "necessitating a maximum floe size of 1 km" should be reconsidered.

*(P16L19-20) "Floe size modeling efforts have focused on the marginal ice zone" I think some citation is needed.

*(P16L30) I could not understand the meaning of "structural uncertainty".

That is all. Faithfully yours.

---

## Short Comment (SC2) · 12 Aug 2019

Estimating the Sea Ice Floe Size Distribution Using Satellite Altimetry: Theory, Climatology, and Model Comparison

The Cryosphere Discussions / August 11, 2019

Hi Chris,

Good idea to use satellite altimeter data to look at sea-ice floe chord lengths, which are related to floe sizes. This allows an Arctic-wide analysis over several years, which has not been done. Here are my (unsolicited) comments, in the nick of time (discussion closes tomorrow)!

Harry Stern (hstern@uw.edu)

Main Comments

1. You reference Rothrock and Thorndike (1984) (R&T hereafter) in several places, which is very appropriate, since that is the fundamental paper on "measuring the sea ice floe size distribution". However, you did not give credit to R&T for the concept of the chord length distribution and its close relationship to the floe size distribution (FSD). This is from the Abstract of R&T:
"Another sampling strategy is to measure the lengths of line segments on floes. The distribution of these chord lengths is equivalent to the distribution of floe diameters."

Furthermore, there is an entire section in R&T called "Chord Length Distribution" in which its relationship to the FSD is derived. Look at R&T equation (4):

$$N(\rho) = \frac{2}{\pi} \int_{c=\rho}^{\infty} (c^2 - \rho^2)^{-1/2} \, dM(c)$$

Now look at your equation (9). The similarity in form, together with the meaning of the variables, must be more than chance. Surely R&T were onto something very similar to what you did. Not to take away from your theoretical development, which possibly goes beyond R&T, but please give proper credit to the originators and developers of the connection between chord length and FSD.

2. In the Abstract, you state: "we produce the first climatology and seasonal cycle of sea ice floe size statistics". However, two previous works also produced a seasonal cycle of floe size statistics, namely Perovich and Jones (2014), which you cite in a different context, and
      Stern, HL, Schweiger, AJ, Stark, M, Zhang, J, Steele, M and Hwang, B. 2018. Seasonal evolution of the sea-ice floe size distribution in the Beaufort and Chukchi seas.
*Elem Sci Anth, 6*: 48. DOI: https://doi.org/10.1525/elementa.305
      See Figure 8 in Stern et al.

3. First paragraph of Section 3 (top of page 8): CryoSat-2 radar echo returns have approximately a constant along-track spacing of 300 meters; floe chords are defined as a continuous sequence of two or more floe echoes; and single isolated floe returns are eliminated. Therefore it seems to

me that the shortest chords must be 600 meters long.  Yet the paper states in multiple places that the analysis applies down to 300 meters.  How is that possible?

4. As you state on page 5 (lines 18-19), the representative radius can represent only those floes whose size is larger than $r_{min}$, the smallest possible floe size sampled.  Agreed.  Furthermore, it seems to me that the representative radius is actually proportional to $r_{min}$.  If $r_{min}$ is halved, the representative radius is halved.  So I don't really understand the use of a representative radius, e.g. as depicted in Figure 3, unless it's to look for changes over time.  The absolute value of the representative radius is simply a reflection of the resolution of CryoSat-2; it doesn't seem to have an intrinsic meaning.  The vast majority of floes in the Arctic are smaller than $r_{min}$.

Minor Comments

Page 5, equation (3).
The inequality $r < D/2$ is backwards.  It should be $r > D/2$.

Page 5, line 10.  For the beta function, use the letter B instead of the Greek $\beta$, and state that B is the beta function, because this is the first place where it appears in the paper.

Page 6, line 10.  I don't understand what is meant by "where we leverage that because it is a probability distribution..."  What is being leveraged?

Page 6, equation (13).  $Rn$ is not defined.  Is it the same thing as $Rn,\varepsilon$ of equation (12)?

Page 7, Figure 2.
(i) In panel (d), the label on the x-axis says "Spacing (m)" but it should be kilometers (km).
(ii) In panel (d), it's impossible to tell which tick mark corresponds to "1 km".  Please use short tick marks for unlabeled values and long tick marks for labeled values.
(iii) The caption says that the satellite track is from January 21, 2014, but the text (page 8, line 17) says January 14, 2018.

Page 8, line 17.  Check date, compare to Fig 2 caption.

Page 8, line 19.  Should "red circle" be "blue circle"?

Page 10, Figure 4.
(i) I don't understand "p" in this figure.  Panel (a) has three curves, two of which have p=0.  Panel (b) has three curves, one with p=0 and one with p=5.  The caption refers to $p < 0.1$ and $p > 0.1$. What is p?
(ii) The x-axes of panels (a) and (b) are labeled "m" (meters?) but should probably be "km".
(iii) The caption (line 3) refers to equation 11 but should probably be equation 13 or 14.
(iv) I don't understand the shading in panels (a) and (b).  The caption says that the gray (or blue) shading is the difference between the blue and black curves.  But where the blue and black curves cross, the difference should be zero, but the shading doesn't reflect that.

Page 11, the "MLE method", and Appendix C. You might mention that the "MLE method" was also recommended and applied by Stern et al. (2018) (On reconciling disparate studies...): see their Section 5.1 for a summary. Your Appendix C through equation C5 is essentially the same as Stern et al. Appendix A.

Page 11, line 16. The range from 300 m to 100 km is not 3 orders of magnitude, it's 2.5. Also, if the smallest chords are in fact 600 m long (see Main Comment #3) then the range would be 2.2 orders of magnitude.

Page 13, lines 18-21. "There is a seasonal cycle in the steepness of the distributional tail: shallowest in early winter and steeper in late winter... the changes across the winter months may be due to a reduction of the largest floes..."
These observations are similar to the ones made by Stern et al. (2018) (Seasonal evolution of...), e.g. in the Abstract: "The mean power-law exponent goes through a seasonal cycle... consistent with the processes of floe break-up in spring followed by preferential melting of smaller floes in summer and the return of larger floes after fall freeze-up." You might add a sentence comparing your results to those of Stern et al.

Page 13, line 30. Change "resolve" to "resolved"

Page 15, line 8. Change "Straits" to "straits" (lower case)

Page 16, line 3. I'd suggest changing "global" to "Arctic-wide" or "pan-Arctic". Also, the use of the phrase "high-resolution" here is highly questionable. Stern et al. (2018) catalogued 18 studies of the FSD. Fifteen of them used higher-resolution data than this study.

Page 16, line 6. Again, I don't think 3 orders of magnitude is accurate.

Page 20, line 4. This sentence doesn't make sense. It should say something like: Take the derivative of $L$ with respect to $\alpha$ and set the result equal to zero to arrive at (equation C4).

Page 20, line 15. There is no $C\alpha$ in Appendix A. Perhaps it should say Equation 10.

Page 20, equation C9. The final summation is missing the natural logarithm function, i.e. it should be the sum over $\ln(D_i/D^*)$.

Page 20, line 22. "...that lie below $D^*$." – should this be "above" $D^*$?

Page 21, line 4. Change "least" to "at least"

Page 21, line 10. Either change "radii" to "radius" or delete the word "a" before "representative".

Page 24, Rothrock and Thorndike (1984) is listed twice.

---

## Author Comment (AC1) · 9 Sep 2019

Dear Dr. Hutchings,

Thanks very much to Thomas Armitage, the anonymous reviewer, and Harry Stern for their careful efforts to help improve our manuscript.

Below we reproduce all reviewers' comments in blue. We respond point-by-point in black text and report changes in indented quotes. At the end of this document is appended a "track changes" version of the revised manuscript.

**Reviewer 1: Dr. Thomas Armitage.**

The paper sets out a framework for estimating geophysical parameters relevant to the sea ice floe size distribution based on floe chord lengths measured by the CryoSat-2 radar altimeter. The paper exploits the fact that it is possible to distinguish between returns originating from leads and returns originating from floes, to estimate chord lengths as the distance over which consecutive CS2 waveforms are identified as floes uninterrupted by leads. These chord lengths are then related to the floe size distribution by statistical considerations. I was asked by the editor to assess the remote sensing aspect of this paper; I have not attempted to assess the theoretical developments because this is outside of my area of expertise, and I leave this to other reviewers. Based on the remote sensing aspect only, I would recommend publication after the authors address the following points.

P1 L8: radio should be radar

We corrected this typo! (pg 1, line 7)

> Applied to the CryoSat-2 radar altimetric record, covering the period from 2010-2018,

I would like to see a little more discussion on the limitations of the technique imposed by the sampling of the radar and its high sensitivity to areas of open water. Any amount of bright open water (lead) in the pulse limited footprint (300m along-track x 1.6km across-track) is likely to be identified as a lead or ambiguous. Also, bright open in the larger beam limited footprint (up to 20km across-track) can potentially lead to an ambiguous return (although this depends on the specific waveform processing). I think this could affect your processing in a few ways:

Thanks for bringing this to our attention.

Your assumption that theta is uniformly sampled: The fact that the altimeter footprint isnt symmetric means that for large theta the footprint is almost tangent to the floe perimeter when they intersect, whereas for small theta the footprint is almost perpendicular to the floe perimeter where they intercept. This means that chords with small theta are more susceptible to snagging or contamination by leads, particularly at the beginning and end of the chord closer to the floe edge. This means that the length of chords with small theta could be underestimated.

We agree chords with small $\theta$ are more susceptible to snagging for footprint orientations that are not entirely "floe". We amended Sec. 3 to read (pg 8, line 4):

A chord length is taken from the midpoint of the first to the midpoint of the last radar echo. Individual chord lengths can be underestimated when continuous floes are separated artificially by producing two or more ambiguous echoes in sequence, or when highly reflective leads dominate the waveform return close to the floe edge and cause measurement dropout (Tilling et al., 2019). Lead contamination, or snagging (Armitage and Davidson, 2014) is more likely when the altimeter cuts of a small section of a floe, i.e. for small values of $\theta$. Overestimates of chord length can also occur when ice floes are in close contact with neighboring floes. Therefore, floe chord lengths should be considered a satellite-derived product, not a true measurement of floe size. The minimum chord length retrieval $D_{min}$ is limited to the CryoSat-2 footprint ( 300 meters along-track) (see the discussion in Appendix B).

In addition, in light of your comments and those of reviewer 2, we have amended the discussion of power law behavior in Section 4, with the analysis starting from a minimum scale $D_{min} = 900$ m (pg 10, line 9). This change did not substantially alter the results presented in the manuscript, for reasons discussed in the response to Dr. Stern below.

We either (a) choose $D^*$ to be 900 m (to reduce the impact of small-size sampling errors discussed in Sec. 2) or (b) use the scheme described in Clauset et al. (2007) to evaluate the most likely value of $D^*$ for a power law tail.

2: Your taking Dmin as 300m (P8 L11): I think the minimum chord length (if you are taking a minimum of two consecutive floe measurements to represent the smallest possible chord) is closer to 600m. This is because for a waveform to be classified as a floe you really expect (at least) the pulse limited footprint to contain entirely sea ice and no open water any amount of open water will dominate the sea ice return. The other side of this is then that you can only assign a range to Dmin (from around 600m up to maybe 1200m if the leads either side are at the forward and backward limit of the along track footprints) and within this range it is ambiguous.

Thanks for pointing this out: we made a typo in describing how a single chord was identified. We now explain (pg 8, line 3):

Floe chords are defined as a continuous sequence of one or more floe echoes, with a gap of one ambiguous echo permitted within a floe sequence to allow for anomalous returns.

and (pg 8, line 12),

However, surface discrimination via altimetry is highly accurate in months without melt ponds, (Peacock and Laxon, 2004; Guerreiro et al., 2017; Quartly et al., 2019), giving confidence that floe echos represent a coherent length of ice. More details on the details of chord identification may be found in Tilling et al. (2019).

As mentioned above, we additionally no longer use $D_{min}$ to classify the power-law behavior across all scales, instead using a more conservative 900 m: (pg 10, line 9),

We either (a) choose $D^*$ to be 900 m (to reduce the impact of small-size sampling errors

discussed in Sec. 2) or (b) use the scheme described in Clauset et al. (2007) to evaluate the most likely value of $D^*$ for a power law tail.

Figure 3 and P9 L3-10: You claim that the largest representative radii lie along the Canadian Archipelago, but the maps show that the largest radii actually surround the pole hole.

(NB: We broke this paragraph up). Thanks, we now changed how we described the region of largest sizes: (pg 9, line 27),

> The largest representative radii in the Arctic lie in the interior Arctic near the pole, with a tongue of large floes that extends along the Canadian Arctic in late winter.

and (pg 14, line 34),

> Geographic variability of representative radius is broadly similar between model and observation: the largest floes lie in the Arctic interior, with regions of smaller floes in the straits and continental margins.

It seems like there is a geophysical signal, with larger radii in the MYI zone, which I would expect, but then superimposed on that is a signal that seems to increase with latitude such that the largest floe sizes are close to the pole hole. I was wondering if you have an explanation for the large floe radii surrounding the pole hole, because this isnt something I would intuitively expect? The spatial sampling increases as you move north due to the convergence of orbits, could this skew the mean floe radius large?

We added a supplementary figure to discuss this: while indeed the representative floe size does increase with latitude, there is not a covariance between floe size and the number of chord measurements. We discuss in the text (pg 9, line 28):

> There is a notable increase of representative radius with latitude. In the Supporting Info Fig. S2, we show that this relationship cannot be explained as a result of the increasing density of measurements near the pole and may therefore be a geophysical signal.

as well as in the Supporting Info, Text S3

> Fig. S2(a) shows the relationship between annual-average representative radius and latitude in the Arctic, which demonstrates a rapid increase above 80°N. Fig. S2(b) shows the relationship between annual-average representative radius and the number of observed chord lengths. There is a weak covariance of the number of chords and representative radius, and therefore we the increase in (a) is not a result of the higher pass density near the pole.

Incidentally I couldnt find Figure S1?

We responded to the reviewer comment with the missing figure and have included it in the new

revision.

**Reviewer 2**

We thank the second reviewer for their efforts in improving our work.

General comments: This paper aims at developing a new method to obtaining the floe size distribution in the Arctic Ocean using the chord length data derived from the CryoSat-2 altimeter data. The conversion of the chord length distribution to floe size distribution statistically seems to the highlight of this paper. The authors accomplished this with a strong mathematical background and an assumption of circular shapes of ice floes and homogeneous and isotropic distribution. With this method, they attempted to show the geographical, seasonal, and interannual properties of the floe size distribution in the Arctic Ocean for the first time. They also tested the validity of power law distribution which has been applied for the floe size distribution frequently. As a result, they concluded that although power law scaling cannot be ruled out, the statistical basis is limited and especially the assumption of power law is weak in the Chukchi and Beaufort seas. They also emphasized the refinement of this method for the operational use.

Floe size distribution is one of the important parameters of sea ice and should be considered to understand the behavior of sea ice area. However, due to the limited observations, its statistical properties are not clarified yet. The idea of using the chord length is interesting in that it can cover wide areas and the truncation error caused by the traditional analysis of finite area seems to be reduced. Therefore, I agree this paper presented an interesting and useful method to analyze the floe size distribution. But to do so, I think the new method should be validated carefully with observational data and the consistency should be confirmed by comparing it with the traditional method. This paper focuses mainly on the availability of the new method with their strong mathematical background. While I feel this is an important step, the validation of this method seems relatively weak. In this case, I think the validation is especially important because their method is based on several assumptions such as circular shapes or homogeneous distribution. Thus, while I agree their method is interesting, my evaluation for this paper is somewhat reserved. In addition, the descriptions are not necessarily readable at places for me, and might be improved, I feel. The major points are as follows:

We thank the reviewer for their careful reading and for placing our study in context of the existing literature. A technically complete validative study is indeed something necessary to pursue, but not within the scope of this study which is to lay out this concept and how it is applied. We explain (pg 3, line 5):

> To date, however, these studies have not been designed to facilitate a comparison with model data, nor have altimetric studies been used to compile floe size statistics. These objectives are the focus of this work.

and (pg 3, line 11):

> One of the key aims of the paper is to develop floe size distribution measurements that are useful for model validation and calibration.

and in the conclusions (pg 17, line 20):

This paper has focused on the framework for making altimetric measurements of the FSD and comparison to model output, but the obtained chord lengths and distributions have not been carefully validated against other observational methods, and this will be necessary before further application of this method.

1) Assumptions of this method. The authors assume that the floe chord distribution data is homogeneous, isotropic, and stationary within the region, and time data is collected (P3L17-18). For simplicity in mathematical treatment, it might be allowed. But I think the validity of these assumptions should be examined somewhere in the paper. For example, fracture patterns caused by shear stress near the shore are far away from circular shapes (e.g., Schulson and Hibler, 1991 Journal of Glaciology). As a preliminary step, I encourage the authors to confirm this method is available even in such situation based on the real floe patterns. Besides, to make an assumption of stationary distribution, the time scale on which this assumption is valid should be discussed because they discuss the seasonal and interannual variation of FSD later. If they examined that their method is applicable to obtain floe size distribution by comparing with the traditional analysis based on real satellite images, the value of this paper would have been enhanced significantly.

We now explain that the timescale of stationarity assumed in this manuscript is one month (pg 9, line 5):

The full CryoSat-2 dataset examined here spans the time period from October 2010 to November 2018, and floe chords measured using the above technique are binned into the CICE sea ice model's two-dimensional sea ice grid for each month and year to facilitate comparison with model products. This implies that the principles of isotropy, homogeneity and stationarity of the FCD, required to produce such a distribution, are invoked on the length scale of the CICE model grid and time scale of a month.

And discuss the issue of different fracture patterns and their impact on statistics on pg 4, line 24,

Nevertheless, it will likely be necessary to amend the analysis below in the future to account for more realistic shape distributions and geometries (e.g., diamonds (Wilchinsky and Feltham, 2006)), regional differences in floe shape properties (such as in regions where shear stress determines ice shape (Schulson and Hibler, 1991)), or to evaluate the sensitivity of the results that follow to the assumed shape distribution.

Theoretical support with mathematics is important, but that is not enough, I think.

This perspective is widely shared by the authors. Far more work is required before this method can be operationally applied to geophysical problems. However, performing a full validation study first requires a theory to validate, and we hope to perform this important work in the coming future.

2) Technical matters To represent floe size, they used a radius of the equivalent circle (P3L7). Is it common? To my knowledge, a diameter has usually been used in the past studies. I think a diameter fits the sense of floe size better, although this might be an essential matter.

We now cite the use of the "effective radius" (pg 3, line 19):

> Define a floes size, $r$, as its "effective radius" — the square root of the floe's area divided by $\pi$ (Rothrock and Thorndike, 1984; Horvat and Tziperman, 2015)) We use radius instead of diameter, as appears in some other observational studies, for comparison with model output in Sec. 5

In the past studies, floe size distribution has been represented in two ways: cumulative number distribution and non-cumulative one. To avoid confusion, it might be better to declare which way this paper takes somewhere in section 2.

We explain the FSD is non-cumulative now when defining the FSD in Sec. 2 (pg 3, line 31),

> In the same region, we define the (non-cumulative) number FSD $P(r)$, where $P(r)dr$ is the fractional number of floes with a size between $r$ and $r + dr$ in $A$, and is also normalized to one.

In section 3, how they determined the individual chord lengths from the Cryosat-2 records is still unclear to me. When ice floes are contacting the neighboring floes, how did they determine the boundary? How much is the measurement accuracy of vertical distance? If it is about 10 cm, it would be quite difficult to identify the edge of the floe for thin ice especially. I think this is a critical matter.

We agree that the measurement of floe boundaries is complicated, as highlighted in the response to Reviewer 1. We now add further clarity in Sec. 3 (pg 8, line 5):

> Individual chord lengths can be underestimated when continuous floes are separated artificially by producing two or more ambiguous echoes in sequence, or when highly reflective leads dominate the waveform return close to the floe edge and cause measurement dropout (Tilling et al., 2019). Lead contamination, or snagging (Armitage and Davidson, 2014) is more likely when the altimeter cuts of a small section of a floe, i.e. for small values of $\theta$. Overestimates of chord length can also occur when ice floes are in close contact with neighboring floes.

And discuss challenges in regions of thin ice in the Discussion (pg 17, line 27):

> The assumption of scale-invariant sampling, observational uncertainty because of the finite sampling resolution, analysis of ambiguous returns, and the accuracy of retrievals in regions of thin sea ice may also affect the inferred size of sea ice floes. This in turn may affect the climatologies described in this study.

In section 4, they tested the validity of power-law distribution. It might be possible that the real FSD may have different regimes although the power-law is applicable for each regime. Figure 4 may suggest such possibility. In such a case, how do they judge the validity?

We explain why we do not consider multiple regimes now (pg 12, line 1):

> We note that a "power law" describes the scaling of a distribution's tail. Previous observational studies have discussed "double power laws" (i.e., Toyota et al., 2011), i.e. two power-law distributions of different exponent joined at a specified scale. The methods employed here would capably capture the large-size power law scaling but not the small-scale scaling. Such "double power laws" are necessarily scale-variant, and require at least 3 parameters to describe. The conceptual and mathematical simplicity of the "power law hypothesis" does not apply in such a case, and we do not consider them here.

Interpretation of the results Overall, I feel the discussion of the results might be a bit weak compared with the preparatory statement about mathematical treatment. For example, they showed During the months of October-December, the climatological representative radius is roughly 35% larger than February-April (P8L32-34). I would like to know how they interpret this result because from intuition floe size tends to become larger at the later growth stage. I am wondering if the measurement accuracy might affect this result significantly.

Thanks, we now explain (pg 9, line 19):

> We interpret this seasonal cycle in size over time as due to the formation of large first-year ice pans in October which are later fractured into smaller floes throughout the winter months

And discuss measurement accuracy in the discussion (pg 17, line 27):

> The assumption of scale-invariant sampling, observational uncertainty because of the finite sampling resolution, analysis of ambiguous returns, and the accuracy of retrievals in regions of thin sea ice may also affect the inferred size of sea ice floes. This in turn may affect the climatologies described in this study.

4) The applications of the results They examined the validity of power-law distribution (section 4) and the application to numerical sea ice models (section 5). But since the validity of this method is not fully investigated from the observations, I am wondering if these applications might be really useful. Especially, it was difficult for me to understand the authors intention about section 5. Personally, at this stage I prefer focusing on showing the validity of this new method to extending the results to models.

We explain the motivation for Sec. 5 in the introduction (pg 3, line 5),

> To date, however, these studies have not been designed to facilitate a comparison with model data, nor have altimetric studies been used to compile floe size statistics. These objectives are the focus of this work. . . .
> One of the key aims of the paper is to develop floe size distribution measurements that are useful for model validation and calibration. In Sec. 5, we show a proof-of-concept,

demonstrating how altimetric data can be used to constrain and evaluate new models of the FSD, comparing the CryoSat-2 FSD data to a climate model simulation with a prognostic FSTD model.

how it may be useful even in the presence of such uncertainty (pg 17, line 33),

Yet observational uncertainties regarding, for example, the floe shape distribution can be roughly estimated at the order of the error in "effective radius" obtained for circular floes ($r = \sqrt{A/\pi}$) or a square ($r = \sqrt{A/4}$), a relative error of 25%. To constrain model results beyond this scale of error will require further refinement. However, as shown in Fig. 6, at present the model-data mismatch in the interior Arctic can exceed a factor of 3. Even with expected levels of error in the present derived FCD/FSD product, some constraints on the model can be considered at present with this method.

and explain the need for validation on pg 17, line 33:

The positive comparison between model and observation in Section 5 could also be due to a compensation between these measurement uncertainties and will need to be re-examined in future validation work.

Specific comments: (P1L2) spare should be replaced by sparse.

Thanks!

(P1L15) covered in sea ice might be covered with sea ice.

Thanks!

(P1L18, P2L12) Rothrock and Thorndike, 1984b I could not see the difference between 1984a and 1984b in the reference lists. If they are the same, please take b.

Indeed there was none, we have fixed!

(P2L19-20) This is true. But to describe this, it would be needed to show that FCD is better than FSD in accuracy.

We would appreciate that this comment be clarified, perhaps the page number is incorrect? We did edit this sentence as follows (pg 2, line 21):

Improvements in the quality and quantity of available FSD data are needed before arriving at consensus derived FSD statistics to guide and assess model performance.

(P3L15) the fraction of floe chords should be the number fraction of floe chords?

Yes, thanks!

For a domain of horizontal area $A$, and over a period of time $\Delta T$ that corresponds to several repeat satellite passes, we bin the set of recorded floe chords to form a probability distribution $S(D)$, which we term the "floe chord distribution" (FCD), where $S(D)dD$ is equal to the number fraction of floe chords in $A$ over $\Delta T$ with length between $D$ and $D + dD$, and is normalized to one.

(P3L3) In Eq.1, F(r;D) S(D) can be replaced by F(r;D) S(D) dD dr to show the number of floes which have radius between r and r+dr. The same applies for the righthand term.

This is true, but for simplicity we hope it is ok to not add differentials to both sides. See the next comment where we improved out explanation of the equation following your comments.

(P3L24) I wonder if F(r;D) dr dD should be F(r;D) S(D) dr dD. (P3L27) Likewise, I wonder if F(D;r) dr dD should be F(D;r) P (r) dr dD.

We have rearranged the sentence to make this more clear (pg 4, line 5)"

[revised manuscript text omitted]

(P16L19-20) Floe size modeling efforts have focused on the marginal ice zone I think some citation is needed.

We changed this to "marginal ice zone processes" and added citation here (pg 17, line 14):

Floe size modeling efforts have focused on marginal ice zone processes (Horvat and Tziperman, 2015; Zhang et al., 2015), and particularly floe sizes below about 1 km because these small floes play an important role in sea ice thermodynamics for floe sizes.

(P16L30) I could not understand the meaning of structural uncertainty.

We changed this to "observational uncertainty" (pg 17, line 27):

The assumption of scale-invariant sampling, and observational uncertainty because of the finite sampling resolution, may also affect the inferred size of sea ice floes.

**Short Comment: Harry Stern**

Hi Chris, Good idea to use satellite altimeter data to look at sea-ice floe chord lengths, which are related to floe sizes. This allows an Arctic-wide analysis over several years, which has not been done. Here are my (unsolicited) comments, in the nick of time (discussion closes tomorrow)!

Harry, thanks very much for taking the time to add your comments here.

**Main Comments**

You reference Rothrock and Thorndike (1984) (R&T hereafter) in several places, which is very appropriate, since that is the fundamental paper on measuring the sea ice floe size distribution. However, you did not give credit to R&T for the concept of the chord length distribution and its close relationship to the floe size distribution (FSD). This is from the Abstract of R&T: Another sampling strategy is to measure the lengths of line segments on floes. The distribution of these chord lengths is equivalent to the distribution of floe diameters. Furthermore, there is an entire section in R&T called Chord Length Distribution in which its relationship to the FSD is derived. Look at R&T equation (4):

Now look at your equation (9). The similarity in form, together with the meaning of the variables, must be more than chance. Surely R&T were onto something very similar to what you did. Not to take away from your theoretical development, which possibly goes beyond R&T, but please give proper credit to the originators and developers of the connection between chord length and FSD.

We appreciate greatly this being brought to our attention — we were (at least consciously) ignorant of the referenced section, but this connection between floe sizes and chords of circular floes was investigated previously, both by R&T and others. We added text to explain this fact (pg 2, line 34):

> One-dimensional measurements of sea ice properties, like along-track altimetric measurements of ice open water, have long been sought to describe the two-dimensional ice surface. Rothrock and Thorndike (1984) originally described a method for reconstructing the sea ice floe size distribution in a region using straight-line measurements over the geometry of floes. Lindsay and Rothrock (1995) later compiled the statistics of lead and ice spacings in two-dimensional imagery. Other work has taken place to derive and understanding the width distribution of individual leads in visual imagery and altimetry (Wadhams et al., 1988; Key and Peckham, 1991; Key, 1993; Wernecke and Kaleschke, 2015), which can be used to estimating heat fluxes and turbulent transfer between the ocean and atmosphere. To date, however, these studies have not been designed to facilitate a comparison with model data, nor have altimetric studies been used to compile floe size statistics. These objectives are the focus of this work.

2. In the Abstract, you state: we produce the first climatology and seasonal cycle of sea ice floe size statistics. However, two previous works also produced a seasonal cycle of floe size statistics, namely Perovich and Jones (2014), which you cite in a different context, and Stern, HL,

Schweiger, AJ, Stark, M, Zhang, J, Steele, M and Hwang, B. 2018. Seasonal evolution of the sea-ice floe size distribution in the Beaufort and Chukchi seas. Elem Sci Anth, 6: 48. DOI: https://doi.org/10.1525/elementa.305 See Figure 8 in Stern et al.

This is a good point, and we should have been more specific. We now change the abstract to state (pg 1 line 7):

> Applied to the CryoSat-2 radar altimetric record, covering the period from 2010-2018, and incorporating 11 million individual floe samples, we produce the first pan-Arctic climatology and seasonal cycle of sea ice floe size statistics.

And in the introduction we explain (pg 2, line 11):

> The observational record of floe statistics derives from visual imagery localized in space and time (i.e., Rothrock and Thorndike, 1984; Toyota et al., 2006; Steer et al., 2008; Toyota et al., 2011) or from repeat measurements in the same region over multiple months (Hwang et al., 2017; Stern et al., 2018a), which may subsequently used to compile a seasonal cycle of the FSD (Perovich and Jones, 2014; Stern et al., 2018a).

3. First paragraph of Section 3 (top of page 8): CryoSat-2 radar echo returns have approximately a constant along-track spacing of 300 meters; floe chords are defined as a continuous sequence of two or more floe echoes; and single isolated floe returns are eliminated. Therefore it seems to me that the shortest chords must be 600 meters long. Yet the paper states in multiple places that the analysis applies down to 300 meters. How is that possible?

As discussed in the response to Dr. Armitage, the "floe chord" dataset includes single echos, and we clarify (pg 8, line 3):

> Floe chords are defined as a continuous sequence of one or more floe echoes, with a gap of one ambiguous echo is permitted within a floe sequence to allow for anomalous returns

As you state on page 5 (lines 18-19), the representative radius can represent only those floes whose size is larger than rmin, the smallest possible floe size sampled. Agreed. Furthermore, it seems to me that the representative radius is actually proportional to rmin. If rmin is halved, the representative radius is halved. So I dont really understand the use of a representative radius, e.g. as depicted in Figure 3, unless its to look for changes over time. The absolute value of the representative radius is simply a reflection of the resolution of CryoSat-2; it doesnt seem to have an intrinsic meaning. The vast majority of floes in the Arctic are smaller than rmin.

The described relationship between representative radius and $r_{min}$ holds under the power-law assumption, with $f(r)$ decaying uniformly and algebraically from $r_{min}$ to infinity. Since we record distributions that have finite maximum values for $r$ and are not power laws, this is not the case: indeed preliminary results from the ICESAT-2 altimeter indicate slightly *larger* floe sizes with a higher resolution. We explain in the text now (pg 6, line 5):

These derived quantities are useful because they require no further information about the sea ice (such as its concentration) to compare against modeled FSDs. However, both $\bar{r}$ and $\mathcal{P}$ can represent only those floes whose size is larger than $r_{min} = D_{min}/2$, the smallest possible floe size sampled. For perfect power-law distributions beginning at a scale of $r_{min}$ or before, both metrics are functions of $r_{min}$. However, for the real FCDs measured here, a maximum floe size exists, and a power-law scaling is not found approaching $r_{min}$, so the use of such metrics is justified (see Sec. 4).

This sensitivity did inform our decision to not produce maps of $\mathcal{P}$, and so we add (pg 6, line 16):

However, because $\mathcal{P}$ is a proportional to a negative moment of the FCD, it is sensitive to changes in the number of small chord lengths. Because of the measurement uncertainty for smaller chord lengths we will focus instead on $\bar{r}$ which is instead a positive moment of the FCD.

**Minor Comments**

Page 5, equation (3). The inequality r ¡ D/2 is backwards. It should be r ¿ D/2.

Thanks! We fixed this inequality.

Page 5, line 10. For the beta function, use the letter B instead of the Greek , and state that B is the beta function, because this is the first place where it appears in the paper.

Indeed, we now state (pg 7, line 3):

$$A_n \equiv \int_0^1 \xi^n G(\xi)d\xi = \frac{2^{n+1}}{\pi} \int_0^{\frac{\pi}{2}} \sin(x)^n dx = \frac{2^n}{\pi} B\left(\frac{n+1}{2}, \frac{1}{2}\right),$$

where $B$ is the beta function.

Page 6, line 10. I dont understand what is meant by where we leverage that because it is a probability distribution... What is being leveraged?

We re-write this (pg 6, line 23):

where the integral of the left-hand side of Eq. 1 is equal to $S(D)$ as $\int F(r; D)dr = 1$

Page 6, equation (13). Rn is not defined. Is it the same thing as Rn, of equation (12)?

Thanks! This equation is now (pg 7, line 7):

$$\alpha_{n,\epsilon} = n + \epsilon \frac{R_{n,\epsilon}}{R_{n,\epsilon} - D_{min}^{\epsilon}} = \text{ constant.} \tag{3}$$

Page 7, Figure 2. (i) In panel (d), the label on the x-axis says Spacing (m) but it should be kilometers (km). (ii) In panel (d), its impossible to tell which tick mark corresponds to 1 km. Please use short tick marks for unlabeled values and long tick marks for labeled values. (iii) The caption says that the satellite track is from January 21, 2014, but the text (page 8, line 17) says January 14, 2018.

Thanks for these edits - we moved the hashes to the plot to avoid this confusion and changed the labeling (see new Fig 2).

Page 8, line 17. Check date, compare to Fig 2 caption.

We fixed this date in the text (pg 9, line 1):

> Figure 2 shows an example of floe chord data for a single CryoSat-2 track over the Arctic on January 21, 2014.

Page 8, line 19. Should red circle be blue circle?

Indeed it should - thanks!

Page 10, Figure 4. (i) I dont understand p in this figure. Panel (a) has three curves, two of which have p=0. Panel (b) has three curves, one with p=0 and one with p=5. The caption refers to p ¡ 0.1 and p ¿ 0.1. What is p? (ii) The x-axes of panels (a) and (b) are labeled m (meters?) but should probably be km. (iii) The caption (line 3) refers to equation 11 but should probably be equation 13 or 14. (iv) I dont understand the shading in panels (a) and (b). The caption says that the gray (or blue) shading is the difference between the blue and black curves. But where the blue and black curves cross, the difference should be zero, but the shading doesnt reflect that.

We altered the legend, fixed the axis labeling and equation, cleared some of the clutter in the image, and eliminated the confusing p-values in the legend. The caption reads (Fig 4, caption):

> Examining the power-law hypothesis. (a) Histogram of all chord lengths recorded in the Arctic for the months November-April (black). Bin centers indicated by hashes and are logarithmically spaced. Blue line is power-law fit to all observed sizes according to eq. 14. Red line is power-law fit to the tail. Dashed red lines are fit lines using the $\pm$ 1 standard deviation values of $\hat{\alpha}$. Red vertical line is the most likely beginning of the power law tail, $D^*$, with shaded region $\pm$ 1 standard deviation in $D^*$. (b) Same as (a), but for measurements in April.

Page 11, the MLE method, and Appendix C. You might mention that the MLE method was also recommended and applied by Stern et al. (2018) (On reconciling disparate studies...): see their Section 5.1 for a summary. Your Appendix C through equation C5 is essentially the same as Stern

Indeed! We now better explain that many studies have been suggesting/deriving/using this method (pg 10, line 5):

> This method has been used to evaluate power-law behavior in recent FSD model Horvat and Tziperman (2017) and observational studies Hwang et al. (2017); Stern et al. (2018b) and proceeds as follows:

During the derivation we now state (pg 21, line 1):

> Following (Muniruzzaman, 1957; Clauset et al., 2009) (see also the derivation in Stern et al. (2018a)),

 The range from 300 m to 100 km is not 3 orders of magnitude, its 2.5. Also, if the smallest chords are in fact 600 m long (see Main Comment #3) then the range would be 2.2 orders of magnitude.

We eliminated the "order of magnitude" statement (pg 12, line 6):

> To illustrate why this is important, we first consider the entire set of 11 million chord lengths recorded in the Arctic in all months (October-April), spanning a length range from 300 m to 100 km.

 There is a seasonal cycle in the steepness of the distributional tail: shallowest in early winter and steeper in late winter... the changes across the winter months may be due to a reduction of the largest floes... These observations are similar to the ones made by Stern et al. (2018) (Seasonal evolution of...), e.g. in the Abstract: The mean power-law exponent goes through a seasonal cycle... consistent with the processes of floe break-up in spring followed by preferential melting of smaller floes in summer and the return of larger floes after fall freeze-up. You might add a sentence comparing your results to those of Stern et al.

We do so now (pg 14, line 10):

> A similar seasonal cycle to that found in Fig. 6(a,c), with an FSD that steepens from September to April, was found in image analysis of floes in the Beaufort and Chukchi Seas (Stern et al., 2018b), with $\alpha \approx 2.5$, although the distribution steepened monotonically over that period.

 Change resolve to resolved
 Change Straits to straits (lower case)

Thanks!

 Id suggest changing global to Arctic-wide or pan-Arctic. Also, the use of the phrase

high-resolution here is highly questionable. Stern et al. (2018) catalogued 18 studies of the FSD. Fifteen of them used higher-resolution data than this study.

Agreed - we made these changes (pg 16, line 30):

> This method provides the first pan-Arctic accounting of climate-relevant quantities derived from the FSD, permits testing of existing scaling laws previously used to characterize distributions of floe size, and allows for gridded comparisons between FSD models and observations.

Page 16, line 6. Again, I dont think 3 orders of magnitude is accurate.

Fixed this (pg 16, line 32):

> Using this new technique we produced climatological, annual-average, and geographic mean moments of the Arctic FSD across a range of resolved length scales from 300 m to 100 km.

Page 20, line 4. This sentence doesnt make sense. It should say something like: Take the derivative of L with respect to  and set the result equal to zero to arrive at (equation C4).

Thanks - we add now (pg 21, line 5):

> As the natural log is monotonically increasing in its argument, to find the most likely $\alpha$, denoted $\hat{\alpha}$, we take the derivative with respect to $\alpha$ and solve a similar equation,

Page 20, line 15. There is no C in Appendix A. Perhaps it should say Equation 10.

We re-organized and re-wrote this material (pg 21, line 10):

> The above analysis concerns the most likely $\alpha$ that explains the FCD. We may ask a separate question: what is the most-likely $\alpha$, which we define as $\alpha_P$, that would explain the FSD, given the explicit relationship that can be derived between $S(D)$ and a power-law distributed $P(r)$ examined in Eq. 10:

Page 20, equation C9. The final summation is missing the natural logarithm function, i.e. it should be the sum over ln (Di/D*).

Thanks, fixed!

Page 20, line 22. ...that lie below D*.  should this be above D*?

We removed this sentence from the revised manuscript.

Fixed, thanks!

Thanks! (pg 22, line 10).

[revised manuscript text omitted]
 include Matlab code that compares the two estimates, and shows that they agree even for small ($N < 25$)

[Figure]

**Figure 2.** Constructing a FCD from altimetry. (a) Base 10 logarithm of the number of floe chords identified, binned into the CESM grid, across all CryoSat returns in the Arctic from 2010-2018. Black line is a single satellite track on January 21, 2014. (b) Subsection of the track centered on the blue dot in (a). Blue line is freeboard of sea ice in radar echoes defined as "floes" along the track. Black lines are chords identified from the freeboard retrieval. (c) Total number of chords measured in each month in the Arctic. Plot is centered on January 1. (d) FCD for the satellite track depicted in (a). Black marks on x-axis are the logarithmically spaced chord length bins.

sets of power-law distributed data.  While in practice Eq 13 is  easy to apply, it only holds when $\alpha_{n,\epsilon} > n + 1$, and unlike the method of Clauset et al. (2009), it does not allow for a robust statistical analysis of the power-law fit, and should only be used when the data is assumed to follow a power-law already.

**3    Climatology and Trends in Floe Properties Derived from CryoSat-2 Altimetry**

We apply the analytic technique described in Sec. 2 to a floe chord data set constructed from the CryoSat-2 radar altimeter processed by the Center for Polar Observation and Modelling (CPOM) over the period from October 2010-present (CPOM data products are available at http://www.cpom.ucl.ac.uk/csopr/seaice.html). CryoSat-2 radar echo returns are defined as "lead", "floe", "open ocean" or "ambiguous"  according to waveform shape and sea ice concentration (Tilling et al., 2016, 2018b), at an approximately constant along-track spacing $D_{min}$ =300 m. Floe chords are defined as a continuous sequence of one or more "floe echoes", with a gap of  one ambiguous echo permitted within a floe sequence to allow for anomalous returns. A chord length is taken from the midpoint of the first to the midpoint

of the last radar echo. Individual chord lengths  can be underestimated when continuous floes are separated artificially by producing two or more ambiguous echoes in sequence. , or when highly reflective leads dominate the waveform return close to the floe edge

5 and cause measurement dropout (Tilling et al., 2019). Lead contamination, or "snagging" (Armitage and Davidson, 2014) is more likely when the altimeter cuts of a small section of a floe, i.e. for small values of $\theta$. Overestimates of chord length can also occur when ice floes are in close contact with neighboring floes. Therefore, floe chord lengths should be considered a satellite-derived product, not a true measurement of floe size. The minimum chord length retrieval $D_{min}$ is limited to the CryoSat-2 footprint ($\sim$ 300 meters along-track) (see the discussion in Appendix B). However, surface discrimination via

10 altimetry is highly accurate in months without melt ponds, (Peacock and Laxon, 2004; Guerreiro et al., 2017; Quartly et al., 2019), giving confidence that  floe echos represent a coherent length of ice. More details on the details of chord identification may be found in Tilling et al. (2019). Indeed, this raw floe chord data has been used successfully to reduce biases in altimeter-observed satellite sea ice thickness estimates from  altimeters with different footprint sizes  Tilling et al. (2019). Here we analyze the sea ice

15 floe size distribution using that floe chord product.

Figure 2 shows an example of floe chord data for a single CryoSat-2 track over the Arctic on January  21, 2014. Freeboard values for echoes discriminated as "floe" are plotted in Fig. 2b as a function of the along-track distance in km, and correspond to the  blue circle in Fig. 2a. Floe chords are identified as black segments in Fig. 2b. The histogram of all 741 identified chords for this single satellite pass is shown in log-log space in Fig. 2d.

20 The full CryoSat-2 dataset examined here spans the time period from October 2010 to November 2018, and floe chords measured using the above technique are binned into the CICE sea ice model's two-dimensional sea ice grid for each month and year to facilitate comparison with model products. This implies that the principles of isotropy, homogeneity and stationarity of the FCD, required to produce such a distribution, are invoked on the length scale of the CICE model grid and time scale of a month. For every grid cell $i$, month $m$, and year $y$, we have a vector of floe chords $\{D_{i,m,y}\}$ from which we build a FCD.

25 The base 10 logarithm of the total number of floe chords recorded in each grid cell per month is shown in Fig. 2a. Because the satellite passes are densest near the pole, the measurement density is highest near the pole as well. Fig. 2c shows the number of Arctic measurements in each month. Sea ice type from CryoSat-2 is not available during summer months, as melt ponds make it difficult to discriminate between leads and ponded floe surfaces, and we do not include measurements from May to September. Across the entire set of satellite tracks included here, 11 million chord lengths are recorded in the Arctic.

30 Figure 3a shows the seasonal cycle of Arctic representative radius over the CryoSat-2 period obtained by applying Eq. 6 to the binned CryoSat-2 floe chord product. Individual years are plotted as thin lines, and the climatological average is shown in red. Details on how temporal and spatial average statistics are computed is included in Appendix D. During the months of October-December, the climatological representative radius is roughly 35% larger (7.06 km vs 5.18 km) than February-April. This seasonal cycle is broadly consistent across years. We interpret this seasonal cycle in size over time as due to the

[revised manuscript text omitted]

---

## Author Comment (AC2) · 9 Sep 2019

Please see the supplement to Dr. Armitage's review in which is attached the full reviewer response.
* * *

---

## Author Comment (AC3) · 9 Sep 2019

Please see the supplement to Dr. Armitage's review in which is attached the full re-viewer response.
* * *

---

## Author Response (AR1)

Dear Dr. Hutchings,

Thanks very much for your efforts here - we respond to your comments below and look forward to working to improve and understand this method in the future.

Line 30 page 4: "where shear stress determines sea ice shape". What shape do you refer to? floe shape? the shape of the ice between leads? Perhaps clarify.

We now clarify:

> . . . in regions where shear stress determines fracture patterns and floe shapes . . .

Line 23 page 7: "aggrement" Spelling.

Thanks - fixed!

Rather than ask the reader to find and interpret the code in the supplimental section, can you provide a brief description of the operations you performed in the paper, referencing the supplimental section.

A good point - we make this more clear:

> In the Supporting Information (Text S1 and File S1), we compare these two estimates when they are evaluated against synthetic datasets drawn from a true power-law distribution. The two agree even when the size of the data is relatively small ($N < 25$).

End of page 9: "We interpret this seasonal cycle in size over time as due to the formation of large first-year ice pans in October which are later fractured into smaller floes throughout the winter months" - Personally I don't like speculation like this. In Autumn storm activity is high, pans of ice can get broken up. I get what you are implying here, that once sheet of ice form and the ocean is calm large pans with grow. Intuitively we would expect leads to crack the entire ice pack once it becomes consolidated to the coast and able to propagate stresses long distances, this is in December/January according to stress measurements taken by Jackie Richter-Menge in the Beaufort in 2001/2002 (Richter-Menge et al. 2003). So there may be some truth in what you say, but it surprises me. It would be best to clarify that further observations are required to understand your results.

We agree speculation of this type isn't great - we added such a comment only at the suggestion of Reviewer 2. To strike a healthier balance we now write:

> A possible interpretation of this seasonal cycle is that large first-year ice pans form in October and are later fractured into smaller floes throughout the winter months. This concept is supported by observations that large-scale fracturing of sea ice in the Beaufort Sea is dominated by coastal processes and therefore only can occur once sea ice freezes

to the coast in mid-winter (Richter-Menge, 2002), although such an interpretation is speculative and must be evaluated further as this method is refined.

line 9 page 14: "With colored cells those where" may read easier as "colored cells are those where"

Thanks - fixed.

line 20, page 14. Consider splitting into two sentences after "disagree". The hyphen confused me on the first read.

Done!

Section 5: Do you consider the larger cut off flow size of 900m that you implimented earlier? This is getting to a scale with relatively few floes in the MIZ though.

We computed means using all FCD measurements to make the comparison as broad as possible - we now explain,

> We include all FCD measurements here (chord lengths above 300 m) to make the broadest comparison, but note that the potential underrepresentation of floes with diameters near the sampling resolution may lead to inaccurate values of $\bar{r}$ in regions mainly consisting of such floes.

I agree with reviewer 2 that this section seams a little out of place in the paper. However you make a reasonable point that the method was developed so models and observations could be compared. Perhaps you can clarify this at the top of the section. It is fair to say that floe size distributions in models is a newly developing field and your method might facilitate testing models following this example. With the caviate of the minimum detectable floe sizing being rather high and further work required to understand seasonality and regional variability.

I think this is important to add as well, we now say:

> A principal aim of this work is to allow model-data comparisons and facilitate testing rapidly-developing FSD/FSTD models. Here we demonstrate how such a comparison can be made and provide useful information to modelers, even in the presence of the high uncertainties in this nascent FSD reconstruction technique.

page 18 around line 5: Can you comment on whether your choice of geographic regions is appropriate to isolate ice regimes that may have power law scaling for floe size? You do show that there is no global power law scaling for the satellite product, and I feel you should be careful to not draw conclusions about local dynamics from this. This does not negate your point that models acounting for floe-scale processes are needed.

Good point - we now mention that there may be issues when zooming in to these small areas -

While we found little evidence for power-law scaling throughout most areas of the Arctic, this may be sensitive to the geographic (here the CICE model grid of approximately 25 km x 25 km) and temporal (here all measurements between 2010-2018) windows we use to collect and evaluate chord length measurements for a power law.

page 18 line 24: "This implies that the principles of isotropy, homogeneity and stationarity of the FCD, required to produce such a distribution, are invoked on the length scale of the CICE model grid and time scale of a month." - define this length scale, because it is not set my CICE itself but by the modellor. Also, the model referenced does not need to be CICE, it can be any model that has a continuum or homogenous element assumption.

We now define the length scale and make clear it is a choice we have made:

This implies that we invoke the principles of isotropy, homogeneity and stationarity of the FCD, required to produce such a distribution, on the length scale of the CICE model grid (O(25 km)) and time scale of a month.

Please make sure you are consistent in your symbols throughout the manuscript, including appendicies. For example, did you change beta consistently throughout?

We checked for this consistency as suggested.

page 22, line 4: Check the brackets.
Equation C10 needs a comma behind it rather than a period.

The line numbers are potentially different on our version vs. yours. We fixed a strange bracket in the bibliography around the word "Arctic", and fixed a period added to Eq. C4.

Reviewer 2 has valid points and I agree further work is needed. I also see that this work will not happen if the method is not published. I encourage you to be explicit in the limitations of the method as you present in this paper, and am looking forward to seeing the work evolve in the future.

We agree entirely, hope we have been sufficiently careful in describing the methodology, and will be working on this a great deal in the months and years to come.